# ASYNCHRONOUS FACTORIZATION FOR MULTI-AGENT REINFORCEMENT LEARNING

## ABSTRACT

Value factorization is widely used to design high-quality, scalable multi-agent reinforcement learning algorithms. However, current methods typically assume agents execute synchronous, 1-step *primitive actions*, failing to capture the typical nature of multi-agent systems. In reality, agents are asynchronous and execute *macro-actions*—extended actions of variable and unknown duration—making decisions at different times. This paper proposes value factorization for asynchronous agents. First, we formalize the requirements for consistency between centralized and decentralized macro-action selection, proving they generalize the primitive case. We then propose update schemes to enable factorization architectures to support macro-actions. We evaluate these asynchronous factorization algorithms on standard macro-action benchmarks, showing they scale and perform well on complex coordination tasks where their synchronous counterparts fail.

## 1 INTRODUCTION

Multi-agent reinforcement learning (MARL) algorithms typically assume that agents have *synchronous* execution (Rashid et al., 2018; Yu et al., 2022; Wang et al., 2021)—each agent selects a *1-step primitive action* that starts and ends simultaneously at each execution step. However, this assumption does not typically hold in real scenarios where agents select and complete actions with varying durations at different times. These temporally extended behaviors are known as *macro-actions* and generalize the primitive case, allowing *asynchronous* execution (Amato et al., 2019). Macro-actions have several advantages over primitive ones as they (i) foster explainability by representing complex multi-step real-world behavior (e.g., navigating to a waypoint, waiting for a human); (ii) benefit value-backup, improving the efficiency of value-based learning (Mcgovern et al., 1999); and (iii) enable action selection to take place at a higher level, using existing controllers to execute behaviors (e.g., a navigation stack) without learning end-to-end actions (e.g., control motor torques). Nonetheless, limited attention has been devoted to this area of research (Jia et al., 2020; Xiao et al., 2022; Liang et al., 2024), motivating the need for principled and scalable approaches.

Due to partial observability and communication constraints, MARL algorithms often learn policies conditioned on local information while leveraging centralized training data to foster collaborative behaviors (i.e., *centralized training with decentralized execution* (CTDE) (Tuyls & Weiss, 2012)). In the synchronous case, value factorization has been successful at CTDE by using a *mixer* network to factor a joint action value $Q_{tot}$ into per-agent utilities conditioned on local information (Rashid et al., 2018; Wang et al., 2021). To achieve a sound factorization, these methods ensure consistency between the local and the joint action selection (i.e., the actions selected from each are the same)—a principle known as the *individual global max* (IGM) (Son et al., 2019). Agents can thus execute in a decentralized manner by selecting actions according to the local utilities while learning in a centralized fashion. Value factorization methods are some of the most scalable and high-performing MARL methods, but extending them to the asynchronous case has yet to be investigated.

This paper introduces value factorization for asynchronous MARL with macro-actions. We lay the theoretical foundations by formalizing *Macro-IGM*—the IGM principle for macro-action-based value functions—and showing it generalizes the primitive case by representing a broader class of functions. On the practical side, we bridge the gap with primitive action-based methods by introducing *asynchronous value factorization* (AVF) algorithms.

Core to AVF is a *macro*-state buffer, which stores extra (state) information conditioned on macro-action selections. Our ablation study shows the importance of such a mechanism, as algorithms trained without it fail to learn good behaviors in simple setups. In contrast to the primitive case, building algorithms on top of Macro-IGM allows us to design different update schemes, as macro-actions can continue or terminate at a certain step (Fig. 1). We propose a *centralized update* that propagates gradient information back to all the agents, regardless of their macro-action execution status. However, in complex set-

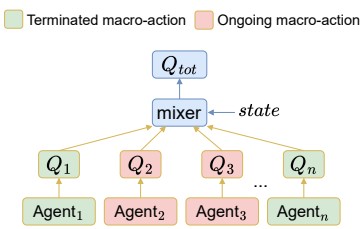

Figure 1: General factorization architecture for asynchronous agents.

tings, we note only some agents might cooperate (i.e., terminating a "coordinated" macro-action at the same time), while others might be involved in other operations. For this reason, we propose two *partially centralized updates* by (i) detaching the gradient of agents with ongoing macro-actions but considering their value when factoring the joint signal; or (ii) masking out (i.e., zeroing) the value of agents with ongoing macro-actions. We expect these update strategies to be more or less effective depending on the task to solve, which we analyze in Section 4 and in our experiments.

We evaluate AVF methods on increasingly complex benchmarks in the macro-action literature (Xiao et al., 2020a;b; 2022). These problems have an increasing number of agents with strict cooperative behaviors to learn, sub-tasks to complete, and severe partial observability. Each one comes with a predefined set of macro-actions; this is the same as assuming primitive actions are given in a primitive task. Our results show that primitive factorization and existing macro-action baselines fail to cope with the complexity of these scenarios. Conversely, AVF methods successfully learn asynchronous decentralized policies in most tasks, allowing us to achieve significantly higher payoffs and learn good decentralized behaviors. *To our knowledge, this is the first formalization of macro-action-based IGM and factorization algorithms.* Our theory and approaches show impressive performance and lay the groundwork for future asynchronous value factorization methods.

## 2 PRELIMINARIES AND RELATED WORK

*Primitive actions* tasks are modeled as *Decentralized Partially Observable Markov Decision Processes* (Dec-POMDPs) (Oliehoek & Amato, 2016) with a tuple $\langle \mathcal{N}, \mathcal{S}, \mathcal{U}, T_s, r, \mathcal{O}, T_{\mathcal{O}}, \gamma \rangle$: $\mathcal{N}, \mathcal{S}$ are finite sets of agents and states; $\mathcal{U} \equiv \langle U^i \rangle_{i \in \mathcal{N}}$ and $\mathcal{O} \equiv \langle O^i \rangle_{i \in \mathcal{N}}$ are the finite sets of primitive joint actions and observations; $U^i, O^i$ are the individual ones. At each step, every agent $i$ picks an action, forming the joint one $\boldsymbol{u} \equiv \langle u^i \in U^i \rangle_{i \in \mathcal{N}}$. After performing $\boldsymbol{u}$, the environment transitions from a state $s$ to a new $s'$, following a transition probability function $T_s : \mathcal{S} \times \mathcal{U} \times \mathcal{S} \to [0, 1]$ (defined as $T_s(s, \boldsymbol{u}, s') = Pr(s'|s, \boldsymbol{u})$), and returning a joint reward $r : \mathcal{S} \times \mathcal{U} \to \mathbb{R}$. Under partial observability, agents receive an observation $\boldsymbol{o} \equiv \langle o^i \rangle_{i \in \mathcal{N}} \in \mathcal{O}$ according to an observation probability function $T_{\mathcal{O}} : \mathcal{O} \times \mathcal{U} \times \mathcal{S} \to [0, 1]$ (defined as $T_{\mathcal{O}}(\boldsymbol{o}, \boldsymbol{u}, s') = P(\boldsymbol{o}|s', \boldsymbol{u})$). Each agent maintains a policy $\pi_i(u_i|h_i)$, mapping local histories $h_i = (o_0^i, u_0^i, \dots, o_t^i) \in H^i$ to actions. In finite-horizon Dec-POMDPs, the objective is to find a joint policy $\pi(\boldsymbol{u}|\boldsymbol{h})$ maximizing the expected discounted return from a state: $V^\pi(s) = \mathbb{E}_\pi \left[ \Sigma_{t=0}^{z-1} \gamma^t r_{t+1} \right]$, where $\gamma \in [0, 1)$ is a discount factor, $z$ is the problem horizon, and $\boldsymbol{h} \in \boldsymbol{H}$ is the joint action-observation history.

### 2.1 VALUE FACTORIZATION

Value factorization methods learn a centralized $Q$-function that is factored over agent utilities and rely on local histories for action selection. Due to their wide adoption in the literature, in the following we describe VDN, QMIX, and QPLEX (Sunehag et al., 2018; Rashid et al., 2018; Wang et al., 2021), primitive factorization methods we use to design the asynchronous algorithms in Section 4.

**Additive (VDN)** factors the joint action-value as a sum of per-agent utilities (Sunehag et al., 2018): $Q(\boldsymbol{h}, \boldsymbol{u}) = \sum_{i=1}^{|\mathcal{N}|} Q_i(h^i, u^i)$ which can only represent a limited set of joint $Q$-functions.

**Monotonic (QMIX)** combines utilities by using a non-linear monotonic mixer that satisfies $\frac{\partial Q(\boldsymbol{h}, \boldsymbol{u})}{\partial Q_i(h^i, u^i)} \geq 0, \forall i \in \mathcal{N}$ (Rashid et al., 2018). The mixer takes extra information as input to better factor $Q(\boldsymbol{h}, \boldsymbol{u})$, using positive weights to enforce monotonicity. QMIX represents a wider range of $Q$-functions than VDN but is still limited to the ones that can be factored into a non-linear monotonic combination. These algorithms are effective CTDE methods as they satisfy the IGM principle

(Son et al., 2019) (Eq. 1). This is particularly important for scalability as it enables tractable joint action selection by deriving the joint greedy action from each agent's local utility. Specifically, the argmax over the joint value function is the same as when argmaxing over each local utility:

$$\arg\max_{\boldsymbol{u}\in\mathcal{U}} Q(\boldsymbol{h},\boldsymbol{u}) = \Big(\arg\max_{u^1\in U^1} Q_1(h^1,u^1),\ \ldots,\arg\max_{u^n\in U^n} Q_n(h^n,u^n)\Big), \forall\,\boldsymbol{h}\in\boldsymbol{H}. \tag{1}$$

**Advantage-based (QPLEX)** uses a decomposition of $Q$-functions to form an equivalent advantage-based IGM (*Adv-IGM*) that requires advantage values to be non-positive. $Q$-functions can be decomposed as the sum of history value and advantage functions as $Q(\boldsymbol{h},\boldsymbol{u}) = V(\boldsymbol{h}) + A(\boldsymbol{h},\boldsymbol{u})$ and QPLEX decomposes learned local $Q_i(h^i,u^i)$ into the following utilities:[1]

$$V(h^i) = \max_{u^i} Q(h^i,u^i) \quad A(h^i,u^i) = Q(h^i,u^i) - V(h^i) \quad \forall i\in\mathcal{N}. \tag{2}$$

Such utilities pass into a transformation module to condition on extra information. Then, QPLEX computes the joint $Q$-function as the above sum, using an attention module to enhance credit assignment (Yang et al., 2020). Crucially, QPLEX's authors show the Adv-IGM can be satisfied by decomposing utilities as Eq. 22 (which limits the range of advantage utilities to be $\leq 0$).

## 2.2 LEARNING MACRO-ACTION-BASED POLICIES

*Macro-Action Dec-POMDPs* (MacDec-POMDPs) (Amato et al., 2019) extend Dec-POMDPs to include durative actions (in addition to the primitive ones) with $\langle\mathcal{M},\hat{\mathcal{O}},T_{\hat{o}^i\in\mathcal{N}}\rangle$: where $\mathcal{M}\equiv\langle M^i\rangle_{i\in\mathcal{N}}$ and $\hat{\mathcal{O}}\equiv\langle\hat{O}^i\rangle_{i\in\mathcal{N}}$ are the set of joint macro-actions and macro-observations. Similar to the primitive case, we define joint macro-action-macro-observation histories (or *macro-histories*) $\hat{\boldsymbol{h}}_t\in\hat{\boldsymbol{H}}$ and local ones $\hat{h}_t^i\in\hat{H}^i$. Macro-actions are based on the *options* framework (Sutton et al., 1999); an agent $i$'s macro-action $m^i$ is defined as a tuple $\langle I_{m^i},\pi_{m^i},\beta_{m^i}\rangle$: $I_{m^i}\subset\hat{H}^i$ is the initiation set; $\pi_{m^i}(\cdot|h^i)$ is the low-level policy associated with the macro-action; and $\beta_{m^i}: H^i\to[0,1]$ is the termination condition.[2] The different histories allow the agents to maintain the necessary information locally to know how to execute and terminate $m^i$. During decentralized execution, agents independently select a macro-action that forms the joint one $\boldsymbol{m}=\langle m^i\rangle_{i\in\mathcal{N}}$, and maintain a high-level policy $\pi_{M^i}(m^i|\hat{h}^i)$. At each step of $m^i$'s low-level policy, agent $i$ independently accumulates the joint reward. Upon terminating its macro-action, an agent $i$ receives a macro-observation $\hat{o}^i\in\hat{O}^i$ according to a macro-observation probability function $T_{\hat{o}^i}: O^i\times M^i\times S\to[0,1]$, defined as $T_{\hat{o}^i}(\hat{o}^i,m^i,s')=Pr(\hat{o}^i|m^i,s')$, and resets the reward accumulation for the next macro-action. The aim is to find a joint high-level policy $\pi_{\mathcal{M}}(\boldsymbol{m}|\hat{\boldsymbol{h}})$ that maximizes the expected discounted return.

### 2.2.1 SYNCHRONOUS AND ASYNCHRONOUS MACRO-ACTION BASELINES.

**Synchronous macro-action MARL.** Early works in the field convert the asynchronous problem into a synchronous one by padding macro-actions to be of equal length, and then solving the resultant Dec-POMDP (Jia et al., 2020). Similarly, Liang et al. (2024) proposes to transform an asynchronous update between temporally extended actions into a primitive 1-step update. Some hierarchical methods have considered learning both macro and primitive actions for cooperative multi-agent settings Tang et al. (2018); Xu et al. (2023). However, as also noted by Tang et al. (2018) and Xiao et al. (2022), they do not address asynchronicity, assuming agents perform macro-actions with the same duration. Hence, these previous works can be viewed as an n-step synchronous MARL version of the primitive case and are unrelated to the asynchronous factorization framework we propose.

**Asynchronous macro-action MARL.** Fully asynchronous centralized and decentralized methods over given macro-actions have also been proposed, and are more closely related to our work (Xiao et al., 2020a;b). In Cen-MADDRQN, a centralized agent maintains a joint macro-history $\hat{\boldsymbol{h}}$, accumulating a joint reward $r(s,\boldsymbol{m},\boldsymbol{\tau}) = \Sigma_{t=t_{\boldsymbol{m}}}^{t_{\boldsymbol{m}}+\boldsymbol{\tau}-1}\gamma^{t-t_{\boldsymbol{m}}}r_t$, where $t_{\boldsymbol{m}}$ is the starting time of $\boldsymbol{m}$, and $t_{\boldsymbol{m}}+\boldsymbol{\tau}-1$ marks its termination when *any* agent finishes a macro-action. Hence, $\boldsymbol{\tau}$ is the number of time steps between any two macro-action terminations. A memory buffer $\mathcal{D}$ is used to store joint transition tuples $\langle\hat{\boldsymbol{o}},\boldsymbol{m},\boldsymbol{m}_-,\hat{\boldsymbol{o}}',r\rangle$. At each training iteration, the centralized agent samples a mini-batch of sequential experiences from $\mathcal{D}$ and filters out the tuples where all the macro-actions are

---

[1]Hence, QPLEX does not learn $V_i(h^i)$, $A_i(h^i,u^i)$ in the agents' networks as in the original dueling architecture (Wang et al., 2016), which can improve performance and sample efficiency.

[2]While we consider a deterministic termination, our results can be trivially extended to a probabilistic one.

still executing. Hence, it updates the centralized value function by minimizing the following loss:

$$\mathbb{E}_{\langle \hat{o}, m, m_-, \hat{o}', r \rangle \sim \mathcal{D}} \left[ \left( r + \gamma^{\tau} Q' \left( \hat{h}', \arg\max_{m'} Q(\hat{h}', m' | m_-) \right) - Q(\hat{h}, m) \right)^2 \right], \tag{3}$$

where $m_-$ is the joint macro-action for agents whose actions will continue at the next step, $Q'$ is a target action-value estimator (van Hasselt et al., 2016). The *conditional prediction* is crucial for a correct estimation as only a few agents typically switch to a new macro-action at the next step. In more detail, the cumulated joint reward is based on any macro-action termination, as often only a few agents terminate their execution at a certain step. As such, estimating a $Q$-value without the conditional operator would imply that all agents will switch to a new macro-action at the next step, making the prediction less accurate and forcing agents to sample a new high-level behavior despite not being done with the previous one (Xiao et al., 2020a). Dec-MADDRQN works similarly to Cen-MADDRQN but learns each agent's Q-function in a decentralized manner. Recently, Policy Gradient (PG) (actor-critic) macro-action algorithms have been proposed (Xiao et al., 2022) but, as shown in our experiments, PG methods can be less sample efficient than value-based ones.

## 3   MACRO-ACTION-BASED IGM

For the primitive and synchronous macro-action cases, the primitive IGM applies. Conversely, we consider an asynchronous setup with macro-actions typically lasting for different, unknown lengths. Hence, to achieve principled asynchronous factorization, we must ensure the consistency of greedy macro-action selection in joint and local macro-action-value functions. The conditional $Q$-values prediction is thus pivotal for correctly formalizing Mac-IGM for asynchronous agents. Broadly speaking, for the joint case, we apply the $\arg\max$ operator on agents sampling a new extended action, while maintaining the same set of ongoing macro-actions. For the decentralized case, only the agents with a terminated macro-action select a new one based on local information. Hence, we have to enforce the macro-action selection consistency on a subset of the agents. This discrepancy, along with conditional joint $Q$-values allows us to adapt existing factorization schemes to asynchronous MARL and design different update schemes leveraging the asynchronous nature of the problem.

**Definition 3.1** (Mac-IGM).  *Given a joint macro-history $\hat{h} \in \hat{H}$, we define the set of macro-action spaces $M^i$ where agent $i$'s macro-action $m^i$ has terminated under local macro-history $\hat{h}^i \in \hat{h}$ as:*

$$\textit{(Terminated macro-action spaces)} \quad \mathcal{M}_+ = \{ M^i \in \mathcal{M} \mid \beta_{m^i \sim \pi_{M^i}(\cdot | \hat{h}^i)} = 1, \forall i \in \mathcal{N} \}. \tag{4}$$

*And define the set of ongoing macro-actions under local macro-history $\hat{h}^i \in \hat{h}$ as:*

$$\textit{(Ongoing macro-actions)} \quad m_- = \{ m^i \in M^i \mid \beta_{m^i \sim \pi_{M^i}(\cdot | \hat{h}^i)} = 0, \ \forall i \in \mathcal{N} \}. \tag{5}$$

*Then, for a joint macro-action-value function $Q : \hat{H} \times \mathcal{M} \mapsto \mathbb{R}^{|\mathcal{M}|}$, if per-agent macro-action-value functions $\langle Q_i : \hat{H}^i \times M^i \mapsto \mathbb{R}^{|M^i|} \rangle_{i \in \mathcal{N}}$ exist such that:*

$$\arg\max_{m \in \mathcal{M}} Q(\hat{h}, m \mid m_-) = \begin{cases} \arg\max_{m^i \in M^i} Q_i(\hat{h}^i, m^i) & \textit{if } M^i \in \mathcal{M}_+ \\ m_-^i & \textit{otherwise} \end{cases} \quad \forall i \in \mathcal{N}, \tag{6}$$

*then, we say $\langle Q_i(\hat{h}^i, m^i) \rangle_{i \in \mathcal{N}}$ satisfies Mac-IGM for $Q(\hat{h}, m \mid m_-)$.*

Definition 3.1 ensures the greedy action selection is the same for both the centralized and decentralized action selection processes only for terminated macro-actions. We can consider a Dec-POMDP to be a degenerate form of a MacDec-POMDP where the macro-actions are primitive actions that terminate after one step. Additionally, primitive actions are included in the macro-action set of each agent: $U^i \subset M^i, \ \forall i \in \mathcal{N}$ (Amato et al., 2014). It follows that Mac-IGM represents a broader class of functions over the primitive IGM. We provide formal proof of such a claim in Appendix A.

**Proposition 3.2.**  *Denoting with*

$$F^{IGM} = \left\{ \left( Q_{IGM} : H \times \mathcal{U} \to \mathbb{R}^{|\mathcal{U}|}, \left\langle Q_{i,IGM} : H^i \times U^i \to \mathbb{R}^{|U^i|} \right\rangle_{i \in \mathcal{N}} \right) \mid \textit{Eq. 1 holds} \right\} \tag{7}$$

$$F^{Mac\text{-}IGM} = \left\{ \left( Q_{Mac\text{-}IGM} : \hat{H} \times \mathcal{M} \to \mathbb{R}^{|\mathcal{M}|}, \left\langle Q_{i,Mac\text{-}IGM} : \hat{H}^i \times M^i \to \mathbb{R}^{|M^i|} \right\rangle_{i \in \mathcal{N}} \right) \mid \textit{Eq. 6 holds} \right\} \tag{8}$$

*the classes of functions satisfying IGM and Mac-IGM respectively, then:*

$$F^{IGM} \subset F^{Mac\text{-}IGM}. \tag{9}$$

Moreover, to design the asynchronous QPLEX algorithm (i.e., AVF-QPLEX), we define the MacAdv-IGM principle that transfers the IGM onto macro-action-based advantage functions.

**Definition 3.3** (MacAdv-IGM). *Given a joint macro-history $\hat{h} \in \hat{H}$, $\mathcal{M}_+$ (Eq. 4) and $m_-$ (Eq. 5) for a joint macro-action-value function $Q : \hat{H} \times \mathcal{M} \mapsto \mathbb{R}^{|\mathcal{M}|}$ defined as $Q_i(h, m|m_-) = V(h) + A_i(h, m|m_-)$, if per-agent macro-action-value functions $\langle Q_i : \hat{H}^i \times M^i \to \mathbb{R}^{|M^i|} \rangle_{i \in \mathcal{N}}$ defined as $Q_i(\hat{h}^i, m^i) = V_i(\hat{h}^i) + A_i(\hat{h}^i, m^i)$ exist such that:*

$$\underset{m \in \mathcal{M}}{\arg\max}\ A(\hat{h}, m \mid m_-) = \begin{cases} \arg\max_{m^i \in M^i} A_i(\hat{h}^i, m^i) & \text{if } M^i \in \mathcal{M}_+ \\ m^i \in m_-^i & \text{otherwise} \end{cases}, \quad \forall i \in \mathcal{N} \qquad (10)$$

*then, we say $\langle Q_i(\hat{h}^i, m^i) \rangle_{i \in \mathcal{N}}$ satisfies MacAdv-IGM for $Q(\hat{h}, m \mid m_-)$.*

Our definition of MacAdv-IGM also differs from the primitive advantage-based IGM (Section 2), since it does require advantage values to be non-positive nor the decomposition of Eq. 22. Nonetheless, it remains an equivalent transformation over the Mac-IGM as shown below. We provide formal proof of such a claim in Appendix A.

**Proposition 3.4.** *The consistency requirement of MacAdv-IGM in Eq. 10 is equivalent to the Mac-IGM one in Eq. 6. Hence, denoting with*

$$F^{\text{MacAdv-IGM}} = \left\{ \left( Q_{\text{MacAdv-IGM}} : \hat{H} \times \mathcal{M} \to \mathbb{R}^{|\mathcal{M}|}, \langle Q_{i,\text{MacAdv-IGM}} : \hat{H}^i \times M^i \to \mathbb{R}^{|M^i|} \rangle_{i \in \mathcal{N}} \right) \mid Eq\ 10\ holds \right\}$$
$$(11)$$

*the class of functions satisfying MacAdv-IGM, we can conclude that $F^{\text{Mac-IGM}} \equiv F^{\text{MacAdv-IGM}}$.*

Similarly to Proposition 3.2, we can also conclude that MacAdv-IGM represents a broader class of functions over the primitive Adv-IGM, and summarize the relationship between the primitive and macro-action classes of functions. Appendix A includes all the missing proofs and discussions.

**Proposition 3.5.** *Denoting with $F^{\{\text{Adv-IGM,MacAdv-IGM}\}}$ the classes of functions satisfying Adv-IGM and MacAdv-IGM, respectively, then:*

$$F^{\text{IGM}} \equiv F^{\text{Adv-IGM}} \subseteq F^{\text{Mac-IGM}} \equiv F^{\text{MacAdv-IGM}}. \qquad (12)$$

# 4 ASYNCHRONOUS VALUE FACTORIZATION

Algorithm 1 presents a general template for our asynchronous factorization approaches, where the centralized network $Q_\Theta$ used during the training phase is composed of agents' decentralized networks $\langle Q_{\theta_i} \rangle_{i \in \mathcal{N}}$, and the chosen mixer module $Q_\phi$. The same holds for the target centralized network typically used in value-based approaches (van Hasselt et al., 2016) (line 2). During decentralized execution (lines 3-15), each agent $i$ maintains an individual local macro-history $\hat{h}^i$ to sample its macro-action $m^i$, and $m^i$'s low-level policy starts its execution at step $t_{m^i}$ and continues until $\beta_{m^i}(h_{t_{m^i}+\tau-1}) = 1$, marking its termination at step $t_{m^i}+\tau-1$ (where $\tau$ is the length of the macro-action). Meanwhile, we accumulate the joint reward signal $r = \Sigma_{t=t_m}^{t_m+\tau-1} \gamma^{t-t_m} r_t$ used to guide the centralized training. Upon terminating its macro-action, agent $i$ receives a new macro-observation $\hat{o}'^i$, macro-state $\hat{s}'^i$, and updates its macro-history $\hat{h}'^i = \langle \hat{h}^i, m^i, \hat{o}'^i \rangle$. Conversely, agents that are still executing their macro-action do not receive new information. We discuss what a macro-state is and its importance in the next section. For centralized training (lines 16-20), the agents use a centralized memory buffer $\mathcal{D}$ to store a joint transition tuple. At each training iteration, we sample

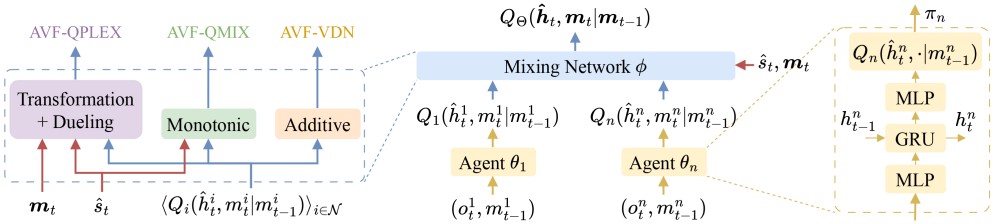

Figure 2: Overview of AVF-based architectures. We summarize the factorization methods we investigate for AVF with purple (QPLEX), green (QMIX), and orange (VDN) boxes.

---

**Algorithm 1** Template for Asynchronous Value Factorization Algorithms

---

1: **Given:** (i) Agents' decentralized and target networks $\langle Q_{\theta_i}, Q_{\theta_i'} \rangle_{i \in \mathcal{N}}$. (ii) Mixer and target mixer networks $Q_\phi, Q_{\phi'}$. (iii) Centralized memory buffer $\mathcal{D}$. (iv) Initial macro observations and macro-states $\langle \hat{o}^i, \hat{s}^i \rangle_{i \in \mathcal{N}}$. (v) Target network update coefficient $\omega$.
2: Define centralized $Q_{\boldsymbol{\Theta}} := (\langle Q_{\theta_i} \rangle_{i \in \mathcal{N}}, Q_\phi)$ and target $Q_{\boldsymbol{\Theta}'} := (\langle Q_{\theta_i'} \rangle_{i \in \mathcal{N}}, Q_{\phi'})$ networks.
3: **while** *training proceeds* **do**
4:     Upon any macro-action termination, reset cumulative reward $r$       *# Decentralized execution*
5:     **for** each agent $i$ **do**
6:         **if** $m^i$ is terminated **then**
7:             Update local history $\hat{h}^i$ with $\hat{o}'^i$ and get the macro-state $\hat{s}^i$ from the environment
8:             $m^i \sim \epsilon$-greedy policy using $Q_{\theta_i}(\hat{h}^i, m^i)$     *# Update info and pick a new macro-action*
9:         **end if**
10:    **end for**
11:    Execute (or continue executing) $\boldsymbol{m} = \{m^i\}_{i \in \mathcal{N}}$ in the environment
12:    Accumulate joint reward $r$
13:    $\hat{o}' \leftarrow \langle \hat{o}'^i \rangle_{i \in \mathcal{N}}$;   $\forall i$, if $m^i$ does not end, $\hat{o}'^i \leftarrow \hat{o}^i$   *# Update upon macro-action termination*
14:    Store the joint transition into $\mathcal{D}$                               *# As Section 4.1*
15:    Sample and filter trajectories as in Section 4.1                *# Centralized training*
16:    Compute per-agent utilities and factorize the joint values
17:    Perform a gradient descent step on $\mathcal{L}(\boldsymbol{\Theta})$ following Eq. 13 on the joint values
18:    Update target weights $\boldsymbol{\Theta}' \leftarrow \omega \boldsymbol{\Theta}' + (1 - \omega) \boldsymbol{\Theta}$
19: **end while**

---

a mini-batch of experiences from this buffer, filtering out the experiences where none of the macro-actions have terminated (Xiao et al., 2020a). We then compute the individual utilities that are fed into the (chosen) mixer, along with the joint macro-state. The mixing network employs the same architecture as the primitive case and outputs the factored joint value driving the learning process. In summary, AVF-based algorithms are trained end-to-end to minimize Eq. 13. After each training step, we update the target weights in a weighted average fashion.

$$\mathbb{E}_{\langle \hat{o}, \hat{s}, \boldsymbol{m}, \boldsymbol{m}_-, \hat{o}', \hat{s}', r \rangle \sim \mathcal{D}} \left[ \left( r + \gamma^\tau Q_{\boldsymbol{\Theta}'} \left( \hat{\boldsymbol{h}}', \arg\max_{\boldsymbol{m}'} Q_{\boldsymbol{\Theta}} \left( \hat{\boldsymbol{h}}', \hat{\boldsymbol{s}}', \boldsymbol{m}' \mid \boldsymbol{m}_- \right) \right) - Q_{\boldsymbol{\Theta}} \left( \hat{\boldsymbol{h}}, \hat{\boldsymbol{s}}, \boldsymbol{m} \right) \right)^2 \right] \quad (13)$$

The overall architecture of AVF-based algorithms is depicted in Fig. 2. On the left, we provide a high-level overview of the primitive value factorization mixers we enable to work in the asynchronous framework (VDN, QMIX, and QPLEX), referring to the resultant algorithms as AVF-{VDN, QMIX, QPLEX}-D0. Overall, these algorithms employ the conditional value functions prediction both in their architecture and update rule, which guarantees to satisfy Mac-IGM and MacAdv-IGM. As a representative example, Appendix B motivates the design of our AVF algorithms by proving the full expressiveness of AVF-QPLEX for MacAdv-IGM.

**Asynchronous updates.** The asynchronous IGM principles also allow us to design different strategies for factoring and updating the agents based on Eq. 13, as depicted in Fig. 3. While the naive D0 performs a "centralized" update propagating gradient information to all the agents (left figure), we propose two "partially centralized" strategies for the only agents with a terminated macro-action by: (i) Masking (i.e., zeroing) the gradient of agents with ongoing macro-actions while considering their value in the mixer, referring to the resultant algorithms with a "D1" suffix (center figure). (ii) Masking the value of agents with ongoing macro-actions in the mixer, referring to the resultant algorithms with a "D2" suffix (right figure).[3] Intuitively, these partially centralized methods should be beneficial in different tasks, depending on their specifications. For example, we expect *D1* methods to perform better when a problem has multiple local optima and requires a specific highly rewarded joint behavior from all the agents (i.e., the higher magnitude of the joint value would incentives the agents to learn such an optimal behavior). Conversely, *D2* methods should offer benefits when a problem comprises several sub-tasks, and only a subset of agents is required to cooperate to solve the sub-tasks. The benchmark tasks employed in Section 5 allow us to investigate these intuitions.

---

[3]Depending on the factorization architecture, masking the value of ongoing agents can result in incorrect value estimation for the agents being updated. Notably, unconstrained mixing architectures that use the joint macro-action history as input are not affected by this issue.

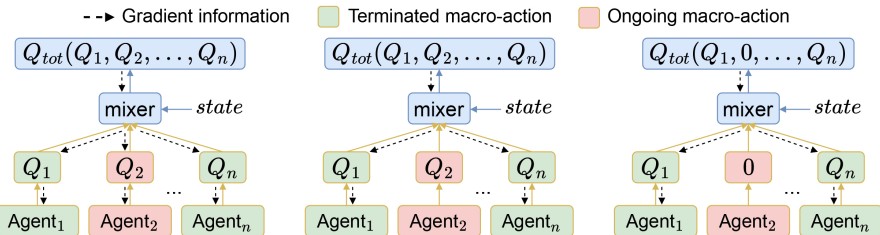

Figure 3: Update schemes for AVF algorithms. (Left) *Centralized (DO):* Agents update despite the status of their macro-action. (Center) *Partially centralized (D1):* Agents with an ongoing macro-action feed their value into the mixer but do not update. (Right) *Partially centralized (D2):* Agents with an ongoing macro-action do not feed their value into the mixer and do not update.

## 4.1 MACRO-STATE BUFFER

The explanatory buffer in Fig. 4 highlights the components affected by (any) macro-action termination with a dashed line at step $t_3$. We train asynchronous agents on samples collected when anyone terminates its macro-action (red boxes), as MacDec-POMDP agents get new data only when their macro-action is done (we discuss a two-agents case, but our example applies to an arbitrary number of agents). In more detail, consider step $t_3$, where agent 1 terminates its macro-action $m_{t_0}^1$ that started at step $t_0$. Upon termination, agent 1 receives a local next observation $o_{t_4}^1$ according to the real state of the environment, $s_{t_4}$ (last column), terminating the joint reward accumulation. The agent thus updates its local macro-history and samples a new macro-action $m_{t_4}^1$. Both the new macro-action and the reward accumulation start in the next step $t_4$. Conversely, agent $n$ does not receive a new observation since its macro-action $m_{t_1}^n$ (started at step $t_1$) has not terminated yet. The state (blue columns) is commonly used as extra information by factorization methods to condition the local utilities and/or the joint value and improve its estimation. In synchronous setups, this is done by simply collecting the state as a features vector at each time step and using it as input for the mixer at centralized training time. However, in the asynchronous case the utilities and joint value are computed over local macro-histories dating back to previous steps and the literature has yet to consider this *temporal inconsistency*. Using the previous example at step $t_3$, we discuss the problem arising from using the environment state and propose two asynchronous alternatives.

**Synchronous case (real state).** The environment transitions to a new state at every step, regardless of macro-action terminations. Using this state is problematic in an asynchronous setup. For example, consider a mixer taking as input individuals' utilities and the environment's state $s$ to estimate the joint value used to update agents. By applying the implicit function theorem (Krantz & Parks, 2002), the joint value can be viewed as a function of individuals' utilities, which let us discuss the temporal inconsistency problem by computing the joint value using the data at step $t_3$:

$$Q\big(\hat{\boldsymbol{h}}_{t_3}, s_{t_3}, \boldsymbol{m}_{t_3} | \boldsymbol{m}_{-t_3}\big) = Q\Big(s_{t_3}, Q_1(\hat{h}_{t_0}^1, m_{t_0}^1), Q_n(\hat{h}_{t_1}^n, m_{t_1}^n)\Big) = Q\Big(Q_1(\hat{h}_{t_0}^1, s_{t_3}, m_{t_0}^1), Q_n(\hat{h}_{t_1}^n, s_{t_3}, m_{t_1}^n)\Big),$$

where $\hat{\boldsymbol{h}}_{t_3} = \langle \hat{h}_{t_0}^1, \hat{h}_{t_1}^n \rangle$, $\boldsymbol{m}_{t_3} = \langle m_{t_0}^1, m_{t_1}^n \rangle$, $\boldsymbol{m}_{-t_3} = \langle m_{t_1}^n \rangle$. Individual utilities are implicitly transformed using $s_{t_3}$, but local histories and macro-actions come from $s_{t_0}, s_{t_1}$. Hence, both agents wrongly condition on $s_{t_3}$. This temporal inconsistency typically leads to high variance and low performance. As a solution, we introduce the notion of a *macro-state*.

**Asynchronous case (macro-state).** Each agent $i$ collects the state of the environment at the time of selecting its macro-action $m_t^i$ (i.e., its macro-state $\hat{s}_t^i$). The agent thus stores a transition to the next

| | Agent 1 | | Agent $n$ | | | |
|---|---|---|---|---|---|---|
| step | transition | state | transition | state | joint reward | real $s$ |
| $t_0$ | $o_{t_0}, m_{t_0}, o_{t_0}$ | $s_{t_0},\ s_{t_0}$ | $o_{t_0}, m_{t_0}, o_{t_1}$ | $s_{t_0},\ s_{t_1}$ | $r_{t_0}$ | $s_{t_0} \to s_{t_1}$ |
| $t_3$ | $o_{t_0}, m_{t_0}, o_{t_4}$ | $s_{t_0},\ s_{t_4}$ | $o_{t_1}, m_{t_1}, o_{t_1}$ | $s_{t_1},\ s_{t_1}$ | $\sum_{k=t_1}^{t_3} \gamma^{k-t_1} r_k$ | $s_{t_3} \to s_{t_4}$ |
| $t_5$ | $o_{t_4}, m_{t_4}, o_{t_4}$ | $s_{t_4},\ s_{t_4}$ | $o_{t_1}, m_{t_1}, o_{t_6}$ | $s_{t_1},\ s_{t_6}$ | $\sum_{k=t_4}^{t_5} \gamma^{k-t_4} r_k$ | $s_{t_5} \to s_{t_6}$ |
| $t_6$ | $o_{t_4}, m_{t_4}, o_{t_7}$ | $s_{t_4},\ s_{t_7}$ | $o_{t_6}, m_{t_6}, o_{t_6}$ | $s_{t_6},\ s_{t_6}$ | $r_{t_6}$ | $s_{t_6} \to s_{t_7}$ |
| $t_9$ | $o_{t_7}, m_{t_7}, o_{t_{10}}$ | $s_{t_7},\ s_{t_{10}}$ | $o_{t_6}, m_{t_6}, o_{t_6}$ | $s_{t_6},\ s_{t_6}$ | $\sum_{k=t_7}^{t_9} \gamma^{k-t_7} r_k$ | $s_{t_9} \to s_{t_{10}}$ |

Figure 4: AVF buffer; green macro-actions continue at the next step; red ones end. We consider different ways to employ extra state information (blue columns).

state when terminating $m_t^i$, similarly to how macro-observations are collected. We identified two ways to input the macro-state in the mixer; we can use: (i) the macro-state of the agent whose macro-action has terminated, or (ii) a joint macro-state comprising the macro-state of all the agents at that step. The former guarantees all the individual utilities with a terminated macro-action transform using the correct (macro-)state. Considering the example at step $t_3$, the first solution leads to:

$$Q(\hat{\boldsymbol{h}}_{t_3}, \hat{s}_{t_0}^1, \boldsymbol{m}_{t_3}|\boldsymbol{m}_{-t_3}) = Q\Big(\hat{s}_{t_0}^1, Q_1(\hat{h}_{t_0}^1, m_{t_0}^1), Q_n(\hat{h}_{t_1}^n, m_{t_1}^n)\Big) = Q\Big(Q_1(\hat{h}_{t_0}^1, \hat{s}_{t_0}^1, m_{t_0}^1), Q_n(\hat{h}_{t_1}^n, \hat{s}_{t_0}^1, m_{t_1}^n)\Big).$$

However, agent $n$ has an ongoing macro-action and transforms its local utility based on temporal inconsistent state information. We propose using the joint macro-state (i.e., $\langle \hat{s}_{t_0}^1, \hat{s}_{t_1}^n \rangle, \rangle$ as input for the mixer to address this issue. We expect the mixer to exploit the only information relevant to each individual, in order to improve the joint estimation. We add an "MS" suffix to AVF algorithms using the joint macro-state. Appendix C discusses the limitations and broader impact of AVF algorithms.

## 5 EMPIRICAL EVALUATION

We aim to answer the following questions: (i) *Can AVF methods learn decentralized policies for complex cooperative tasks? How do different update schemes perform?* (ii) *Do AVF algorithms improve performance over their primitive versions and existing asynchronous macro-action baselines (Dec-MADDRQN, Cen-MADDRQN, Mac-IAICC (Xiao et al., 2020a; 2022)) and a synchronous one (HAVEN (Xu et al., 2023))* (iii) *Are the claims on temporal inconsistency (i.e., the relevance of the macro-state) supported by empirical evidence?* All the algorithms are run over 20 seeds, and data are collected on Xeon E5-2650 CPU nodes with 64GB of RAM, using the hyper-parameters discussed in Appendix D. Appendix E also discusses the environmental impact of our experiments.

We use standard benchmark environments in the macro-action literature (Appendix F): (i) *BoxPushing* (BP). The goal is to move the big box to the goal. An agent can push the small box, but the big one requires both agents to push it simultaneously. Agents only observe the state of the cell in front of them, making high-dimensional grids hard. We consider BP-{10, 30}, where the number indicates the size of the grid. (ii) *Warehouse Tool Delivery* (WTD). A continuous space with multiple workers assembling an item. Four phases are required to complete the item, and one requires a tool. The manipulator searches for the right tool and handles it to the mobile robots, which have to deliver it to the worker. Agents must learn the correct tools for each phase, observing the workstation's state only when close to it. We consider four variants. WTD-S: one working human and two mobile robots. WTD-D: two working humans with one faster work phase and two mobile robots. WTD-T: three working humans with different speeds and three mobile robots. WTD-F: four working humans that work at a fixed speed and three mobile robots. (iii) *Capture Target* (CT). A group of agents has to capture a randomly moving target simultaneously. When successful, agents get a reward of 1. Agents observe their position and the correct target's location with probability 0.3. We significantly increased the complexity of the original CT by considering 10 agents and 1 target.

### 5.1 AVF EXPERIMENTS

Tab. 1 reports the average return and standard error at convergence—our experiments consider a total of 15 AVF algorithms over 7 environments, which does not allow us to visualize the complete training curves for the over 2000 training runs (included in Appendix G, along with more visually friendly bar plots). We remind D0 is the naive centralized update, D1 masks the gradient for agents with an ongoing macro-action (but not their value in the mixer), and D2 masks their values with a 0. Algorithms employing the macro-state of terminated agents (or no extra information) are AVF-{VDN, QMIX, QPLEX}-{D0, D1, D2}, and the ones using the joint macro-state are AVF-{QMIX, QPLEX}-{D0, D1, D2}-MS. Notably, each environment has different characteristics influencing the performance of our update schemes. For example, BP has two agents with very limited observations, and the optimal behavior involves a specific joint action (i.e., both agents have to push the big box simultaneously), but there are other positively rewarded sub-optimal behaviors (e.g., push the individual boxes). In OSD, only a subset of agents are required to cooperate at a certain step (e.g., the manipulator can only deliver one object at a time, to one mobile robot), while others are either waiting or delivering items to (non-learning) humans. Finally, CT has the highest number of agents that have to reach a flickering target at the same time, and no sub-optimal are positively rewarded. As such, we expect different update schemes to have widely different performance across the tasks.

Table 1: Average return and standard error at convergence for all our algorithm variations–tasks have different characteristics affecting the performance of the different update schemes.

| | BP-10 | BP-30 | WTD-S | WTD-F | CT |
|---|---|---|---|---|---|
| AVF-VDN-D0 | $298.8 \pm 0.3$ | $271.4 \pm 0.5$ | $\mathbf{262.0 \pm 4.5}$ | $\mathbf{1049.1 \pm 21.2}$ | $0.00 \pm 0.09$ |
| AVF-VDN-D1 | $\mathbf{298.8 \pm 0.2}$ | $\mathbf{298.8 \pm 0.3}$ | $256.8 \pm 3.1$ | $-243.5 \pm 47.6$ | $0.61 \pm 0.05$ |
| AVF-VDN-D2 | $298.8 \pm 0.3$ | $\mathbf{298.8 \pm 0.3}$ | $253.4 \pm 4.2$ | $843.2 \pm 29.9$ | $\mathbf{0.64 \pm 0.06}$ |
| AVF-QMIX-D0 | $38.9 \pm 4.1$ | $32.9 \pm 7.1$ | $\mathbf{261.6 \pm 2.7}$ | $909.4 \pm 26.4$ | $0.64 \pm 0.05$ |
| AVF-QMIX-D1 | $131.0 \pm 2.2$ | $39.3 \pm 7.8$ | $-130.0 \pm 17.5$ | $-213.3 \pm 24.1$ | $0.70 \pm 0.05$ |
| AVF-QMIX-D2 | $89.8 \pm 4.7$ | $70.3 \pm 6.5$ | $-213.2 \pm 38.2$ | $909.7 \pm 25.8$ | $0.68 \pm 0.07$ |
| AVF-QMIX-D0-MS | $\mathbf{298.8 \pm 0.2}$ | $160.7 \pm 3.4$ | $256.8 \pm 5.1$ | $919.6 \pm 19.1$ | $0.00 \pm 0.01$ |
| AVF-QMIX-D1-MS | $298.8 \pm 0.3$ | $\mathbf{298.8 \pm 0.5}$ | $47.2 \pm 51.1$ | $34.1 \pm 51.6$ | $\mathbf{0.73 \pm 0.03}$ |
| AVF-QMIX-D2-MS | $28.9 \pm 6.9$ | $102.1 \pm 3.9$ | $248.8 \pm 6.6$ | $\mathbf{966.8 \pm 21.5}$ | $0.54 \pm 0.02$ |
| AVF-QPLEX-D0 | $187.9 \pm 29.6$ | $33.45 \pm 9.8$ | $243.8 \pm 6.1$ | $510.3 \pm 17.6$ | $0.10 \pm 0.06$ |
| AVF-QPLEX-D1 | $32.7 \pm 23.3$ | $21.1 \pm 9.1$ | $-148.0 \pm 14.7$ | $-240.3 \pm 23.4$ | $0.73 \pm 0.04$ |
| AVF-QPLEX-D2 | $-10.0 \pm 5.1$ | $-0.36 \pm 3.4$ | $246.3 \pm 5.7$ | $870.7 \pm 24.6$ | $0.61 \pm 0.04$ |
| AVF-QPLEX-D0-MS | $\mathbf{298.8 \pm 0.1}$ | $235.9 \pm 9.9$ | $\mathbf{256.8 \pm 3.7}$ | $553.6 \pm 58.9$ | $0.03 \pm 0.03$ |
| AVF-QPLEX-D1-MS | $\mathbf{298.8 \pm 0.1}$ | $\mathbf{298.8 \pm 0.4}$ | $69.6 \pm 54.0$ | $-42.8 \pm 50.5$ | $\mathbf{0.76 \pm 0.04}$ |
| AVF-QPLEX-D2-MS | $43.7 \pm 42.6$ | $-10.0 \pm 3.1$ | $-68.0 \pm 74.4$ | $\mathbf{918.5 \pm 17.8}$ | $0.39 \pm 0.02$ |

**Overall performance.** *Among centrally updated methods* (D0), *AVF-VDN-D0 has the highest overall performance* but fails to cope with the complex CT task. Both AVF-{QMIX, QPLEX}-D0 fail to learn the joint behavior required by the BP domain, *but AVF-QMIX-D0 is superior to AVF-QPLEX-D0 in all the other tasks*. These results are interesting since, in the primitive case, the overall ranking between VDN, QMIX, and QPLEX is usually the opposite. We motivate this difference as macro-actions drastically simplify the horizon (i.e., number of actions) required to solve problems, and the less complex architectures are more suitable to learn quicker from shorter horizons.

**Comparing different updates.** Considering the partially centralized schemes (D1, D2), we note different trends. *In BP, AVF-{VDN, QMIX}-{D1, D2} obtain higher performance* than their D0 counterparts, but the same does not hold for the AVF-QPLEX versions. *In WTD tasks, the gradient masking of ongoing agents (D1) is detrimental to performance* since only a subset of agents are "actively" cooperating. In contrast, masking the values of ongoing agents (D2) has comparable performance for AVF-VDN, while appearing to be slightly beneficial for AVF-QMIX in the most complex variations of the task. Similarly, AVF-QPLEX-D2 has higher performance than the other update schemes. Finally, *both the partially centralized schemes (D1, D2) significantly outperform the centralized ones (D0) in the CT task* for all these AVF algorithms.[4]

**Joint macro-state.** Using the macro-state of the terminated agents (as analyzed so far) possibly lead to temporal inconsistency. Here, we analyze how using the joint macro-state in the mixer (i.e., MS methods) impacts performance. *When comparing the same algorithm and update scheme, the joint macro-state leads to a significant overall performance improvement.* On top of that, we note in some specific settings (AVF-{QMIX, QPLEX}-{D0, D1}-MS in BP, AVF-QMIX-D0-MS in WTD-{S, D} tasks), MS algorithms achieve high performance, while their macro-state version fail.

**Takeaways.** Overall, methods using the joint macro-state (MS) have higher performance than others under any update scheme, supporting our claims on the importance of temporal consistency. Moreover, each update scheme leads to better performance in specific tasks, suggesting they are all viable but distinct solutions to tackle the challenges of asynchronous MARL.

## 5.2 ADDITIONAL COMPARISONS AND ABLATION STUDY

**Macro-action.** We compare our methods with Dec-MADDRQN Cen-MADDRQN, Mac-IAICC, and HAVEN using their original implementations. Tab. 2 shows the results achieved by these baselines in the most complex tasks. We note AVF algorithms achieve the best performance in all the domains, confirming the benefits of asynchronous MARL over fixed-length macro-actions.

---

[4]D2 masking does not present significant performance drawbacks despite the potential incorrect estimation caused by the value masking.

Table 2: Average return a nd standard error at convergence for previous asynchronous MARL baselines and a hierarchical macro-action-based approach (HAVEN).

|        | Mac-IAICC          | Dec-MADDRQN      | Cen-MADDRQN       | HAVEN             |
| ------ | ------------------ | ---------------- | ----------------- | ----------------- |
| BP-30  | $39.1 \pm 26.9$    | $62.3 \pm 31.3$  | $\mathbf{270.5 \pm 4.1}$ | $188.2 \pm 8.5$   |
| WTD-F  | $\mathbf{900.8 \pm 23.3}$ | $-295.6 \pm 39.6$ | $405.4 \pm 20.4$  | $410.9 \pm 24.1$  |
| CT     | $0.29 \pm 0.06$    | $0.05 \pm 0.05$  | $\mathbf{0.35 \pm 0.02}$ | $0.34 \pm 0.05$   |

Table 3: Average return and standard error at convergence for primitive factorization VDN, QMIX, and QPLEX in the primitive version of the tasks.

|        | VDN              | QMIX             | QPLEX            |
| ------ | ---------------- | ---------------- | ---------------- |
| BP-10  | $-4.1 \pm 10.5$  | $-2.0 \pm 9.3$   | $-13.4 \pm 2.6$  |
| WTD-S  | $-32.5 \pm 4.5$  | $-44.3 \pm 6.2$  | $-61.0 \pm 5.3$  |
| CT     | $0.04 \pm 0.01$  | $0.09 \pm 0.03$  | $0.20 \pm 0.04$  |

**Primitive action.** Tab. 3 reports the performance of the primitive VDN, QMIX, and QPLEX in primitive BP-10, WTD-S, CT (described in Appendix F). Overall, the 1-step algorithms struggle to cope with the complexity of these tasks, since they require a high degree of cooperation, and are characterized by significant partial observability.

**State ablation and Mac-IGM relevance.** Figure 5 shows the issues of using the environment state and the significance of Mac-IGM in representative tasks BP-10 and WTD-S. To investigate the effect of using the environment state, we replaced the joint macro-state (MS) with the raw environment state (last column of Fig. 4), referring to these variants as {QMIX, QPLEX}-D0-S. These variations yielded significantly lower returns, whereas MS-based methods effectively solved the tasks. This result supports our hypothesis that temporally uncorrelated data hinders the learning of high-performing, joint asynchronous policies. To evaluate the role of Mac-IGM, we removed conditional $Q$-value prediction from the AVF algorithms, causing agents to select a new macro-action whenever any macro-action terminated. These variations are referred to as the unconditioned {QMIX, QPLEX}-D0-UC. Consistent with previous findings (Xiao et al., 2020a), unconditioned functions introduced high variability in value estimations, ultimately preventing agents from solving even the easiest BP and WTD tasks. These experiments emphasize the importance of our AVF algorithm design, which incorporates conditional operators and leverages the macro-state effectively.

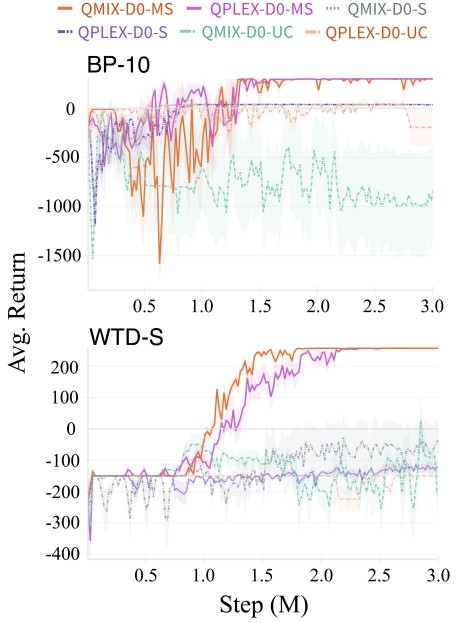

Figure 5: Results for AVF algorithms ("AVF" is omitted for simplicity) using the environment (S), against the joint macro-state (MS).

## 6 CONCLUSION

This paper introduces value factorization for asynchronous MARL to design scalable macro-action algorithms. To this end, we proposed the IGM principle for macro-actions, ensuring consistency between centralized and decentralized greedy action selection. In addition, we showed the proposed Mac-IGM and MacAdv-IGM paradigms generalize the primitive ones and represent a wider class of functions. We also introduced AVF algorithms that leverage asynchronous decision-making and value factorization, under multiple update schemes. Our approach relies on a joint macro-state to maintain temporal consistency in local agents' state information, allowing the use of existing factorization architectures. Crucially, the proposed AVF framework can be applied with arbitrary mixing strategies. Overall, our methods successfully learn asynchronous decentralized policies for challenging tasks where primitive factorization and previous macro-action methods perform poorly.

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

APPENDICES

# A  REPRESENTATIONAL COMPLEXITY OF MAC-IGM AND MACADV-IGM

As discussed by VDN and QMIX (Sunehag et al., 2018; Rashid et al., 2018), common value factorization approaches cannot guarantee representing their respective classes of true value functions in a Dec-POMDP. The same limitation holds in MacDec-POMDPs; agents' observations do not represent the full state in partially observable settings. Similarly, per-agent value function ordering can (potentially) be wrong in a macro-action context. Formally, given an agent $i$ at a time step $t$ it could happen that:

$$Q_i(\hat{h}^i, m^i) > Q_i(\hat{h}^i, m'^i) \text{ when } Q(s, (\boldsymbol{m}^{-i}, m^i)) < Q\left(s, (\boldsymbol{m}^{-i}, m'^i)\right)$$

where $\boldsymbol{m}^{-a}$ is the joint action of all the agents excluding $i$. However, there are several ways to alleviate such an issue. First, it is possible to condition per-agent action values (or state and advantage values) with state information during offline training as in QMIX (Rashid et al., 2018), QPLEX (Wang et al., 2021). Moreover, if can not assume that $(\hat{\boldsymbol{h}}, \boldsymbol{m})$ (i.e., the joint macro-history and action) is sufficient to fully model $Q(s, \boldsymbol{m})$ (which is a common assumption in prior factorization approaches), we can potentially store additional history-related information in recurrent layers (Sunehag et al., 2018).

## A.1  REPRESENTATIONAL EXPRESSIVENESS OF AVF ALGORITHMS

The proposed AVF framework does not change the architectural design of the chosen factorization method. Hence, the algorithms investigated in Section 5, namely AVF-{VDN, QMIX, QPLEX}, maintain the same considerations of the original factorization methods in terms of representational expressiveness.

In particular, AVF-VDN can factorize arbitrary joint macro-action value functions that can be additively decomposed into individual utilities. AVF-QMIX extends the family of factorizable functions to non-linear monotonic combinations. Finally, AVF-QPLEX does not involve architectural constraints and is capable of achieving the entire class of functions satisfying the underlying IGM.

### A.1.1  OMITTED PROOFS IN SECTION 3

**Proposition 3.2.** *Denoting with*

$$F^{IGM} = \left\{ \left( Q^{IGM} : \boldsymbol{H} \times \mathcal{U} \to \mathbb{R}^{|\mathcal{U}|}, \left\langle Q_i^{IGM} : H^i \times U^i \to \mathbb{R}^{|U^i|} \right\rangle_{i \in \mathcal{N}} \right) \mid Eq. \ 1 \ holds \right\} \tag{14}$$

$$F^{Mac\text{-}IGM} = \left\{ \left( Q^{Mac\text{-}IGM} : \hat{\boldsymbol{H}} \times \mathcal{M} \to \mathbb{R}^{|\mathcal{M}|}, \left\langle Q_i^{Mac\text{-}IGM} : \hat{H}^i \times M^i \to \mathbb{R}^{|M^i|} \right\rangle_{i \in \mathcal{N}} \right) \mid Eq. \ 6 \ holds \right\} \tag{15}$$

*the class of functions satisfying IGM and Mac-IGM respectively, then:*
$$F^{IGM} \subset F^{Mac\text{-}IGM} \tag{16}$$

*Proof.* MacDec-POMDPs extends Dec-POMDPs by replacing the primitive actions available to each agent with option-based macro-actions. However, as shown in (Amato et al., 2019), the macro-action set contains primitive actions to guarantee the same globally optimal policy:
$$U^i \subset M^i, \ \forall i \in \mathcal{N} \tag{17}$$
Meaning that $\forall i \in \mathcal{N}, \ |M_i| > |U_i|$, which implies $|\mathcal{M}| > |\mathcal{U}|$. It also follows that $\mathcal{O} \subseteq \hat{\mathcal{O}}$ as a MacDec-POMDP is, in the limit where only primitive actions are selected, equivalent to a Dec-POMDP. For these reasons, we can conclude that $|\hat{\boldsymbol{H}} \times \mathcal{M}| > |\boldsymbol{H} \times \mathcal{U}|$ (i.e., the domain over which primitive action-value functions are defined is smaller than the domain over which macro-action-value functions are defined). Hence, $F^{IGM} \subset F^{Mac\text{-}IGM}$. $\qquad\square$

**Proposition 3.4.** *The consistency requirement of MacAdv-IGM in Eq. 10 is equivalent to the Mac-IGM one in Eq. 6. Hence, denoting with*

$$F^{MacAdv\text{-}IGM} = \left\{ \left( Q^{MacAdv\text{-}IGM} : \hat{\boldsymbol{H}} \times \mathcal{M} \to \mathbb{R}^{|\mathcal{M}|}, \langle Q_i^{MacAdv\text{-}IGM} : \hat{H}^i \times M^i \to \mathbb{R}^{|M^i|} \rangle_{i \in \mathcal{N}} \right) \mid Eq \ 10 \ holds \right\} \tag{18}$$

*the class of functions satisfying MacAdv-IGM, we can conclude that $F^{Mac\text{-}IGM} \equiv F^{MacAdv\text{-}IGM}$.*

*Proof.* Given a joint macro-history $\hat{\boldsymbol{h}} \in \hat{\boldsymbol{H}}$ on which $\langle Q_i(\hat{h}^i, m^i) \rangle_{i \in \mathcal{N}}$ satisfies Mac-IGM for $Q(\hat{\boldsymbol{h}}, \boldsymbol{m} \mid \boldsymbol{m}_-)$, we show Eq. 10 represents the same consistency constraint as Eq. 6. By applying the dueling decomposition from (Wang et al., 2016), we know $Q(\hat{\boldsymbol{h}}, \boldsymbol{m} \mid \boldsymbol{m}_-) = V(\hat{\boldsymbol{h}}) + A(\hat{\boldsymbol{h}}, \boldsymbol{m} \mid \boldsymbol{m}_-)$, and $Q_i(\hat{h}^i, m^i) = V(\hat{h}^i) + A_i(\hat{h}^i, m^i)$, $\forall i \in \mathcal{N}$. Hence, the state-value functions defined over macro-histories do not influence the action selection process. For the joint value, we can thus conclude that:

$$\arg\max_{\boldsymbol{m} \in \mathcal{M}} Q(\hat{\boldsymbol{h}}, \boldsymbol{m} \mid \boldsymbol{m}_-) = \arg\max_{\boldsymbol{m} \in \mathcal{M}} V(\hat{\boldsymbol{h}}) + A(\hat{\boldsymbol{h}}, \boldsymbol{m} \mid \boldsymbol{m}_-) = \arg\max_{\boldsymbol{m} \in \mathcal{M}} A(\hat{\boldsymbol{h}}, \boldsymbol{m} \mid \boldsymbol{m}_-) \quad (19)$$

Similarly, for the individual values:

$$\forall i \in \mathcal{N}, \begin{cases} \arg\max_{m^i \in M^i} Q_i(\hat{h}^i, m^i) & \text{if } M^i \in \mathcal{M}_+ \\ \boldsymbol{m}_-^i & \text{otherwise} \end{cases}$$
$$= \begin{cases} \arg\max_{m^i \in M^i} V(\hat{h}^i) + A_i(\hat{h}^i, m^i) & \text{if } M^i \in \mathcal{M}_+ \\ \boldsymbol{m}_-^i & \text{otherwise} \end{cases} \quad (20)$$
$$= \begin{cases} \arg\max_{m^i \in M^i} A_i(\hat{h}^i, m^i) & \text{if } M^i \in \mathcal{M}_+ \\ \boldsymbol{m}_-^i & \text{otherwise} \end{cases}$$

Broadly speaking, we know the history values act as a constant for both the joint and local estimation and do not influence the $\arg\max$ operator. By combining Eq. 19, 20, we conclude the equivalence between Eq. 6, 10. $\qquad \square$

**Proposition 3.5.** *Denoting with $F^{\{Adv\text{-}IGM, MacAdv\text{-}IGM\}}$ the classes of functions satisfying Adv-IGM and MacAdv-IGM, respectively, then:*

$$F^{IGM} \equiv F^{Adv\text{-}IGM} \subset F^{Mac\text{-}IGM} \equiv F^{MacAdv\text{-}IGM}. \quad (21)$$

*Proof.* The result naturally follows from Proposition 3.2, 3.5, and the result of (Wang et al., 2021) that showed the equivalence between the class of functions represented by the primitive IGM and Adv-IGM. In more detail, from the latter we know $F^{\text{IGM}} \equiv F^{\text{Adv-IGM}}$. Moreover, Proposition 3.2 showed us that $F^{\text{IGM}} \subset F^{Mac\text{-}IGM}$, from which follows that $F^{\text{Adv-IGM}} \subset F^{Mac\text{-}IGM}$. In addition, Proposition 3.5 showed us that $F^{Mac\text{-}IGM} \equiv F^{\text{MacAdv-IGM}}$. Combining these results, we conclude the relationship in Eq. 21. $\qquad \square$

# B   EXPRESSIVENESS OF AVF-QPLEX-D0

In this section, we show how the design of AVF algorithms allows the underlying factorization architecture to maintain the same class of expressiveness as their primitive counterparts (e.g., additive functions, monotonic functions), but with respect to Mac-IGM. Let us prove the full expressiveness AVF-QPLEX-D0 over Mac-IGM as an explanatory example, extending the full expressiveness of QPLEX over IGM of the primitive case.

**Proposition 3.5.** *Given the universal function approximation of neural networks, the function class that AVF-QPLEX-D0 can realize is equivalent to what is induced by Mac-IGM.*

*Proof.* The proof extends the synchronous, primitive action proof of Wang et al. (2021). The main difference is related to the conditional action-value functions learned by AVF-QPLEX-D0, which allows it to maintain action selection consistency and correct updates over asynchronous macro-action-based agents.

First, let us define the utilities deriving from the transformation and mixer modules of AVF-QPLEX-D0. For clarity, we recall these components implement the same operations as the original QPLEX (shown in Fig. 6), but in the asynchronous macro-actions context.

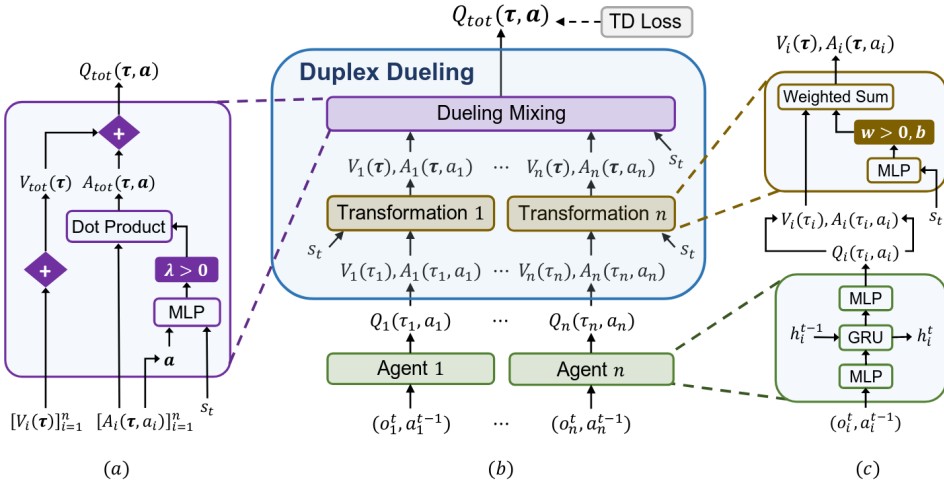

Figure 6: Primitive actions-based QPLEX architecture (image credit: Wang et al. (2021)). (a) Mixing network; (b) QPLEX architecture; (c) Individual utility and transformation networks.

At any step $t$, consider the set of terminated macro-action spaces $\mathcal{M}_{t,+}$ and the ongoing macro actions $m_{t,-}$ defined as in Def. 3.1. For each agent $i$, AVF-QPLEX-D0 first decomposes its utility $Q_i(\hat{h}_t^i, m_t^i | m_{t-1}^i)$ as follows:

$$V_i(\hat{h}_t^i) = \begin{cases} \max_{m^i} Q_i(\hat{h}_t^i, m^i) & if\ M^i \in \mathcal{M}_{+,t} \\ Q_i(\hat{h}_t^i, m_{t-1}^i) & otherwise \end{cases}, \tag{22}$$

$$A_i^{(}\hat{h}_t^i, m_t^i | m_{t-1}^i) = Q_i(\hat{h}_t^i, m_t^i | m_{t-1}^i) - V_i(\hat{h}_t^i).$$

The transformation module then outputs the following transformed utilities:

$$V_i^T(\hat{\boldsymbol{h}}_t) = w_i(\hat{\boldsymbol{h}}_t) V_i(\hat{h}_t^i) + b_i(\hat{\boldsymbol{h}}_t), \tag{23}$$
$$A_i^T(\hat{\boldsymbol{h}}_t, m_t^i | m_{t-1}^i) = w_i(\hat{\boldsymbol{h}}_t) A_i(\hat{h}_t^i, m_t^i | m_{t-1}^i),$$

and the mixer module combines all the agents' utilities into the following joint utilities:

$$V^{MIX}(\hat{\boldsymbol{h}}_t) = \sum_{i \in \mathcal{N}} V_i^T(\hat{\boldsymbol{h}}_t), \tag{24}$$
$$A^{MIX}(\hat{\boldsymbol{h}}_t, \boldsymbol{m}_t | \boldsymbol{m}_{t,-}) = \sum_{i \in \mathcal{N}} \lambda_i(\hat{\boldsymbol{h}}_t, \boldsymbol{m}_t | \boldsymbol{m}_{t,-}) A_i^T(\hat{\boldsymbol{h}}_t, m_t^i | m_{t-1}^i),$$

to finally output the joint value $Q(\hat{\boldsymbol{h}}_t, \boldsymbol{m}_t | \boldsymbol{m}_{t,-})$ defined as:

$$Q^{MIX}(\hat{\boldsymbol{h}}_t, \boldsymbol{m}_t | \boldsymbol{m}_{t,-}) = V^{MIX}(\hat{\boldsymbol{h}}_t) + A^{MIX}(\hat{\boldsymbol{h}}_t, \boldsymbol{m}_t | \boldsymbol{m}_{t,-}). \tag{25}$$

We can now prove the full expressiveness of AVF-QPLEX-D0 over Mac-IGM. Assume AVF-QPLEX-D0's network size is sufficient to satisfy the universal function approximation theorem (Csáji, 2001). Denote the joint $Q^{MIX}, A^{MIX}, V^{MIX}$, transformed $Q_i^T, A_i^T, V_i^T$, and individual $Q_i, A_i, V_i$ macro-action, macro-observation, advantage macro-history-based value functions and utilities learned by AVF-QPLEX-D0, respectively. Moreover, Let the class of action-value functions that the algorithms can represent be $\mathcal{Q}^{MIX}$ defined as:

$$\mathcal{Q}^{MIX} = \left\{ \left( Q^{MIX}, \langle Q_i \rangle_{i \in \mathcal{N}} \right) | \text{Eqs.} 22, 23, 24, 25 \text{are satisfied} \right\}, \tag{26}$$

and let $\mathcal{Q}^{Mac\text{-}IGM}$ be the class of macro-action-value functions represented by Mac-IGM (Eq. 15).

Firstly, we note the multiplicative weights in both the transformation and mixer modules are all positive to satisfy action selection consistency. Secondly, we prove $\mathcal{Q}^{MIX} = \mathcal{Q}^{Mac\text{-}IGM}$ by demonstrating the inclusion in the two directions $\mathcal{Q}^{Mac\text{-}IGM} \subseteq \mathcal{Q}^{MIX}$ and $\mathcal{Q}^{Mac\text{-}IGM} \supseteq \mathcal{Q}^{MIX}$.

1. $Q^{Mac\text{-}IGM} \subseteq Q^{MIX}$: For any $\left(Q^{Mac\text{-}IGM}, \langle Q_i^{Mac\text{-}IGM} \rangle_{i \in \mathcal{N}}\right) \in Q^{Mac\text{-}IGM}$ we construct $Q^{MIX} = Q^{Mac\text{-}IGM}$ and $\langle Q_i \rangle_{i \in \mathcal{N}} = \langle Q_i^{Mac\text{-}IGM} \rangle_{i \in \mathcal{N}}$, deriving $A_i, V_i, A^{MIX}, V^{MIX}$ by Eqs. 22, 24 and constructing the transformed values connecting joint and individual ones as:

$$Q_i^T(\hat{\boldsymbol{h}}_t, \boldsymbol{m}_t | \boldsymbol{m}_{t,-}) = \frac{Q^{MIX}(\hat{\boldsymbol{h}}_t, \boldsymbol{m}_t | \boldsymbol{m}_{t,-})}{|\mathcal{N}|},$$

$$V_i^T(\hat{\boldsymbol{h}}_t) = \max_{\boldsymbol{m}'} Q_i^T(\hat{\boldsymbol{h}}_t, \boldsymbol{m}' | \boldsymbol{m}_{t,-}), \quad A_i^T(\hat{\boldsymbol{h}}_t, \boldsymbol{m}_t | \boldsymbol{m}_{t,-}) = Q_i^T(\hat{\boldsymbol{h}}_t, \boldsymbol{m}_t | \boldsymbol{m}_{t,-}) - V_i^T(\hat{\boldsymbol{h}}_t).$$

According to the fact that $\forall \boldsymbol{m}^* \in \mathcal{M}^*(\hat{\boldsymbol{h}}), \boldsymbol{m} \in \mathcal{M} \setminus \mathcal{M}^*(\hat{\boldsymbol{h}}), i \in \mathcal{N}$:

$$A^{MIX}(\hat{\boldsymbol{h}}, \boldsymbol{m}^* | \boldsymbol{m}_-) = A_i(\hat{h}^i, m^{i,*} | \boldsymbol{m}_-^i) = 0,$$

$$A^{MIX}(\hat{\boldsymbol{h}}, \boldsymbol{m} | \boldsymbol{m}_-) < 0, A_i(\hat{h}^i, m^i | \boldsymbol{m}_-^i) < 0,$$

where $\mathcal{M}^*(\hat{\boldsymbol{h}}) = \{\boldsymbol{m} | \boldsymbol{m} \in \mathcal{M}, Q^{MIX}(\hat{\boldsymbol{h}}, \boldsymbol{m} | \boldsymbol{m}_-) = V^{MIX}(\hat{\boldsymbol{h}})\}$, and by setting:

$$w_i(\hat{\boldsymbol{h}}) = 1, \quad b_i(\hat{\boldsymbol{h}}) = V_i^T(\hat{\boldsymbol{h}}) - V_i(\hat{h}_i),$$

$$\lambda_i(\hat{\boldsymbol{h}}_t, \boldsymbol{m}_t | \boldsymbol{m}_{t,-}) = \begin{cases} \frac{A_i^T(\hat{\boldsymbol{h}}_t, \boldsymbol{m}_t | \boldsymbol{m}_{t,-})}{A_i(\hat{h}_t^i, m_t^i | \boldsymbol{m}_{t,-}^i)} & \text{if } A_i(\hat{h}_t^i, m_t^i | \boldsymbol{m}_{t,-}^i) < 0, \\ 1 & \text{otherwise.} \end{cases}$$

we conclude that $\left(Q^{MIX}, \langle Q_i \rangle_{i \in \mathcal{N}}\right) \in Q^{Mac\text{-}IGM}$, meaning that $Q^{Mac\text{-}IGM} \subseteq Q^{MIX}$.

2. $Q^{Mac\text{-}IGM} \supseteq Q^{MIX}$: For any $\left(Q^{MIX}, \langle Q_i \rangle_{i \in \mathcal{N}}\right) \in Q^{MIX}$, following the above fact regarding non-positive advantage functions/utilities, $\forall \hat{\boldsymbol{h}} \in \hat{\boldsymbol{H}}, i \in \mathcal{N}$, let:

$$A_i^{MIX^*}(\hat{h}^i) = \{m^i | m^i \in \mathcal{M}^i, A_i(\hat{h}^i, m^i | \boldsymbol{m}_-^i) = 0\}.$$

Combining the positivity of the weights $\langle w_i, \lambda_i \rangle_{i \in \mathcal{N}}$ with Eqs. 22, 23, 24, 25, we can derive $\forall \hat{\boldsymbol{h}} \in \hat{\boldsymbol{H}}, m^{i,*} \in A_i^{MIX^*}(\hat{h}^i), m^i \in \mathcal{M} \setminus A_i^{MIX^*}(\hat{h}^i), i \in \mathcal{N}$:

$$A_i(\hat{h}^i, m^{i,*} | \boldsymbol{m}_-^i) = 0 \quad \text{and} \quad A_i(\hat{h}^i, m^i | \boldsymbol{m}_-^i) < 0$$

$$A_i^T(\hat{\boldsymbol{h}}, m^{i,*} | \boldsymbol{m}_-^i) = w_i(\hat{\boldsymbol{h}}) A_i(\hat{h}^i, m^{i,*} | \boldsymbol{m}_-^i) = 0 \quad \text{and}$$

$$A_i^T(\hat{\boldsymbol{h}}, m^i | \boldsymbol{m}_-^i) = w_i(\hat{\boldsymbol{h}}) A_i(\hat{h}^i, m^i | \boldsymbol{m}_-^i) < 0$$

$$A^{MIX}(\hat{\boldsymbol{h}}, \boldsymbol{m}^* | \boldsymbol{m}_-) = \lambda_i(\hat{\boldsymbol{h}}, \boldsymbol{m}^* | \boldsymbol{m}_-) A_i^T(\hat{\boldsymbol{h}}, m^{i,*} | \boldsymbol{m}_-^i) = 0 \quad \text{and}$$

$$A^{MIX}(\hat{\boldsymbol{h}}, \boldsymbol{m} | \boldsymbol{m}_-) = \lambda_i(\hat{\boldsymbol{h}}, \boldsymbol{m} | \boldsymbol{m}_-) A_i^T(\hat{\boldsymbol{h}}, m^i | \boldsymbol{m}_-^i) < 0.$$

Following the proof of (Wang et al., 2021), we can thus construct $Q^{MIX} = Q^{Mac\text{-}IGM}, \langle Q_i \rangle_{i \in \mathcal{N}} = \langle Q_i^{Mac\text{-}IGM} \rangle_{i \in \mathcal{N}}$, meaning that $\left(Q^{Mac\text{-}IGM}, \langle Q_i^{Mac\text{-}IGM} \rangle_{i \in \mathcal{N}}\right) \in Q^{MIX}$, and $Q^{MIX} \subseteq Q^{Mac\text{-}IGM}$.

Under the assumption that AVF-QPLEX-D0's neural networks provide universal function approximation, the joint macro-action-value function class that AVF-QPLEX-D0 can represent is thus equivalent to what is induced by Mac-IGM. $\qquad\square$

## C  LIMITATIONS AND BROADER IMPACT

**Limitations.** We identify three limitations in our work. First, most factorization approaches cannot guarantee to fully represent their respective classes of value functions in a Dec-POMDP (Sunehag et al., 2018; Rashid et al., 2018; 2020); the same limitation holds in AVF-based algorithms that maintain the same representation expressiveness of the original methods. Second, AVF methods employing the joint macro-state could have scalability issues when considering many agents. While such a problem does not arise in our experiments with up to 10 agents, it is possible to train an encoder to reduce the dimensionality of the joint macro-state. Third, MacDec-POMDPs assume

that macro-actions are known and fixed. This is the same as assuming primitive actions are given in a primitive MARL domain. Moreover, asynchronous settings are common in the real world but have been rarely studied in the MARL literature. For this reason, principled methods are needed for the MacDec-POMDP case before extending them to learn macro-actions (e.g., by employing skill discovery approaches (Eysenbach et al., 2019)).

**Broader impact.** Regarding the broader impact of our work, we do believe macro-actions have the potential to scale MARL into the real world. Temporally extended actions enable decision-making at a higher level and naturally represent complex real-world behavior (e.g., lifting an object). that can exploit existing robust controllers or be defined by a (human) expert, making them more explainable than other sequences of primitive actions. By extending MARL algorithms to the macro-action case, realistic multi-agent coordination problems can be solved that are orders of magnitude larger than problems solved by previous primitive MARL algorithms.

## D  HYPER-PARAMETERS

Regarding the considered baselines, we employed the original authors' implementations and parameters (Sunehag et al., 2018; Rashid et al., 2018; Wang et al., 2021; Xiao et al., 2020a; 2022). Table 4 lists all the hyper-parameters considered in our initial grid search for tuning the algorithms employed in Section 5. We separate algorithm-specific parameters (e.g., for the mixer of AVF-QMIX, AVF-QPLEX) with a horizontal line at the end of the table. We tested different joint reward schemes for macro-actions (e.g., only considering the max/min values and time horizon among the agents, averaging them). Still, the original joint scheme in Section 2.2 resulted in the best performance.

Table 4: Hyper-parameters candidate for initial grid search tuning.

| | |
|---|---|
| Learning rate | 5e-4, 2.5e-4, 2.5e-5 |
| $\gamma$ | 0.9, 0.95, 0.99 |
| ASVB (full episodes) size | 1000, 2500, 5000 |
| Batch size | 32, 64, 128 |
| Sampling trajectory size | 10, 25, 50 |
| Polyak averaging $\omega$ | 0.995, 0.9998 |
| N° hidden layers | 2, 3 |
| Hidden layers size | 64, 128 |
| Mix embed. size | 32, 64 |
| Hypernet embed. size | 32, 64 |
| N° hypernet layers | 2 |
| N° Advantage hypernet layers | 2 |
| Advantage hypernet embed. size | 32, 64 |

Table 5 lists the hyper-parameters considered in our experiments. When a parameter differs from the algorithm variations and environments, we indicate the values with a separator. Shared parameters between all the algorithms are indicated once.

## E  ENVIRONMENTAL IMPACT

Despite each training run being "relatively" computationally inexpensive due to the use of CPUs, the experiments of our evaluation led to cumulative environmental impacts due to computations that run on computer clusters for an extended time. Nonetheless, it is crucial to foster sample efficiency (i.e., reducing the training time for the agents, hence the computational resources used to train them) to reduce the environmental footprint of such learning systems. In this direction, our work considers designing macro-action methods that significantly improve the sample efficiency of the learning algorithms (i.e., the number of simulation steps required to learn a policy), as shown by previous research on the topic (Xiao et al., 2020a;b).

Our experiments were conducted using a private infrastructure with a carbon efficiency of $\approx 0.275 \frac{\text{kgCO}_2\text{eq}}{\text{kWh}}$, requiring a cumulative $\approx$360 hours of computation. Total emissions are estimated

Table 5: Hyper-parameters used in our experiments (considering all the algorithm variations).

|  | AVF-VDN | AVF-QMIX | AVF-QPLEX |
|---|---|---|---|
| Learning rate | 5e-4 — 2.5e-4 | 5e-4 — 2.5e-4 — 2.5e-5 | 5e-5 — 2.5e-4 — 2.5e-5 |
| $\gamma$ | | 0.9 | |
| ASCB size | | 2500 | |
| Batch size | | 32 — 64 | |
| Sampling traj. size | | 10 — 25 | |
| $\omega$ | | 0.995 | |
| N° hidden layers | | 2 | |
| Hidden layers size | | 64 | |
| Mix embed. size | - | 32 | 32 |
| Hypernet embed. size | - | 32 | - |
| N° hypernet. layers | - | 2 | - |
| N° Adv. hypernet layers | - | - | 2 |
| Adv. hypernet embed. size | - | - | 32 |

to be $\approx 10.39\text{kgCO}_2\text{eq}$ using the Machine Learning Impact calculator, and we purchased offsets for this amount through Treedom.

# F    DOMAIN DESCRIPTION

## F.1    BOX PUSHING (BP)

In this collaborative task, two agents have to work together to push a big box to a goal area at the top of a grid world to obtain a higher credit than pushing the small box on each own. The small box is movable with a single agent, while the big one requires two agents to push it simultaneously.

The state space consists of each agent's position and orientation, as well as the location of each box. Agents have a set of primitive actions, including *moving forward*, *turning left* or *right*, and *staying* in place. The available macro-actions are **Go-to-Small-Box(i)** and **Go-to-Big-Box** that navigates the agent to a predefined waypoint (red) under the corresponding box and terminates with a pose facing it; and a **Push** macro-action that makes the agent move forward and terminate when the robot hits the world boundary or the big box. Each agent observation is very limited in both the primitive and macro level, which is the state of the front cell: empty, teammate, boundary, small box, or big box.

The team receives a terminal reward of $+300$ for pushing the big box to the goal area or $+20$ for pushing one small box to the goal area. If any agent hits the world's boundary or pushes the big box on its own, a penalty of $-10$ is issued. An episode terminates when any box is moved to the goal area or reaches the maximum horizon, 100 time steps. In our work, we consider the variant of this task in terms of the grid world size as shown in Fig. 9.

The original work of Xiao et al. (2020a) also released a primitive action version of the BP task. In the primitive action version, each agent has four actions: move forward, turn left, turn right, and stay. The small box moves forward one grid cell when any robot faces it and executes the move *move forward* action.

## F.2    WAREHOUSE TOOL DELIVERY (WTD)

Warehouse Tool Delivery scenarios vary in the number of agents, humans, and the speed at which they work. In each scenario, the humans assemble an item with four work phases. Each phase requires several primitive time steps and a specific tool. We assume that the human already holds the tool for the first phase, and the rest must be found and delivered in a particular order by a team of robots to finish the subsequent work phases. The objective of the robot team is to assist the humans in completing their tasks as quickly as possible by finding and delivering the correct tools in the proper order and timely fashion without making the humans wait.

The environmental space is continuous, and the global state includes 1) each mobile robot's 2D position; 2) the execution status of the manipulator robot's macro-action in terms of the rest of primitive time steps to terminate; 3) the work phase of each human with its completed percentage; and 4) each tool's position.

Mobile robots have three navigation macro-actions: 1) *Go-W(i)* moves the robot to the corresponding workshop and locates at the red spot in the end; 2) *Go-TR* leads the robot to the red waypoint in the middle of the tool room; 3) *Get-Tool* navigates the robot the pre-allocated waypoint beside the manipulator and wait there, which will not terminate until either receiving a tool or waiting there for 10 time steps. Mobile robots move at a fixed velocity and are only allowed to receive tools from the manipulator rather than the human. There are three applicable macro-actions for the manipulator robot: 1) *Search-Tool(i)* takes 6 time steps to find a particular tool and place it in a staging area when there are less than two tools there; otherwise, it freezes the robot for the same amount of time.; 2) *Pass-to-M(i)* takes 4 time steps to pick up the first found tool from the staging area and pass it to a mobile robot; 3) *Wait-M* consumes 1 time step to wait for a mobile robot.

Each mobile robot is always aware of its location and the type of tool carried by itself. Meanwhile, it is also allowed to observe the number of tools in the staging area or a human's current work phase when it is at the tool room or the corresponding workshop, respectively. The macro-observation of the manipulator robot is limited to the type of tools present in the staging area and the identity of the mobile robot waiting at the adjacent waypoints.

Rewards for this domain are structured such that the team earns a reward of $+100$ when they deliver a correct tool to a human on time. However, if the delivery is delayed, an additional penalty of $-20$ is imposed. Moreover, the team incurs a penalty of $-10$ if the manipulator robot attempts to pass a tool to a mobile robot that is not adjacent, and a penalty of $-1$ happens every time step.

We consider four variations of WTD shown in Fig. 8: a) WTD-S, involves one human and two mobile robots; b) WTD-D, involves two humans and two mobile robots; c) WTD-T, involves three humans and two mobile robots. d) WTD-F, involves four humans and three mobile robots. The human working speeds under different scenarios are listed in Table 6

Table 6: The number of time steps each human takes on each working phase in scenarios.

| Scenarios | WTD-S | WTD-D | WTD-T | WTD-F |
|---|---|---|---|---|
| Human-0 | $[20, 20, 20, 20]$ | $[27, 20, 20, 20]$ | $[38, 38, 38, 38]$ | $[40, 40, 40, 40]$ |
| Human-1 | N/A | $[27, 20, 20, 20]$ | $[38, 38, 38, 38]$ | $[40, 40, 40, 40]$ |
| Human-2 | N/A | N/A | $[27, 27, 27, 27]$ | $[40, 40, 40, 40]$ |
| Human-3 | N/A | N/A | N/A | $[40, 40, 40, 40]$ |

Each episode stops when all humans have obtained the correct tools for all work phases or when the maximum time steps (150) are reached.

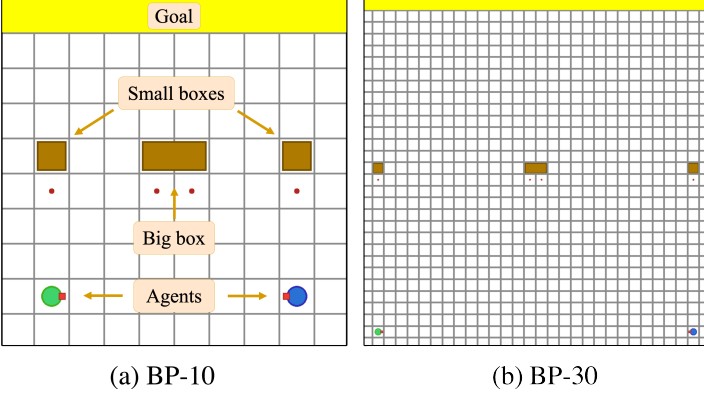

(a) BP-10            (b) BP-30

Figure 7: Overview of the considered box pushing task variations.

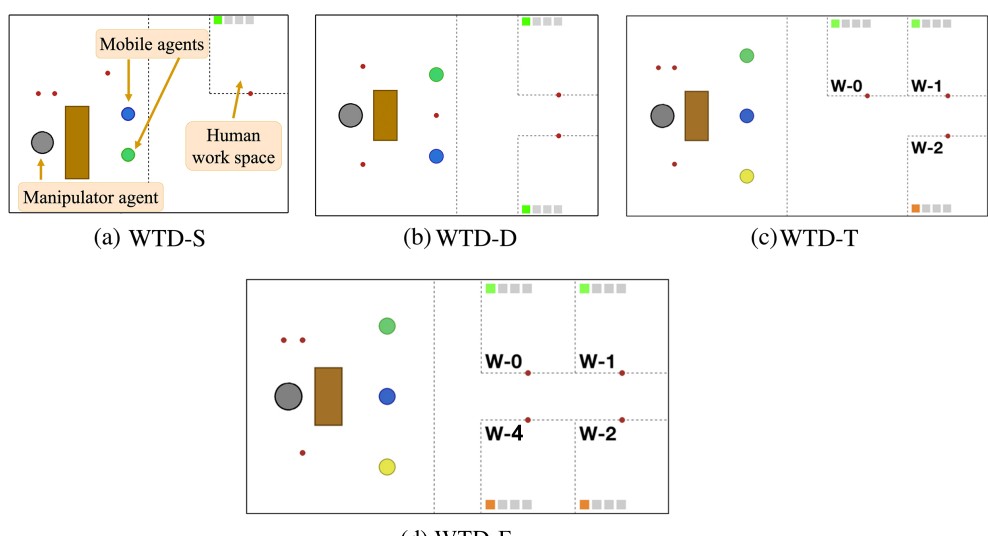

Figure 8: Overview of the considered warehouse tool delivery task variations.

## F.3 CAPTURE TARGET (CT)

In this domain, there are 10 agents represented by blue circles, assigned with the task of capturing a randomly moving target indicated by a red cross (as shown in Fig. **??**). Each agent's macro-observation captures the same information as its primitive one, including the agent's position (being always observable) and the target's position (being partially observable with a flickering probability of 0.3). The applicable primitive-actions include moving *up*, *down*, *left*, *right*, and *stay*. The macro-action set consists of **Move-to-T**, directs the agent to move towards the target with an updated target position according to the latest primitive observation, and *Stay* lasts a single time step. The horizon of this task is 60 time steps, and a terminal reward of $+1$ is given only when all agents capture the target simultaneously by being in the same cell.

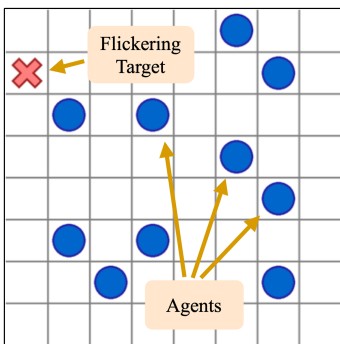

Figure 9: Overview of the considered capture target task.

## G    MISSING PLOTS FROM SECTION 5

For a clearer visualization of the results in Table 1, Figure 10 shows the normalized average return at convergence for all our algorithm variations and environments.

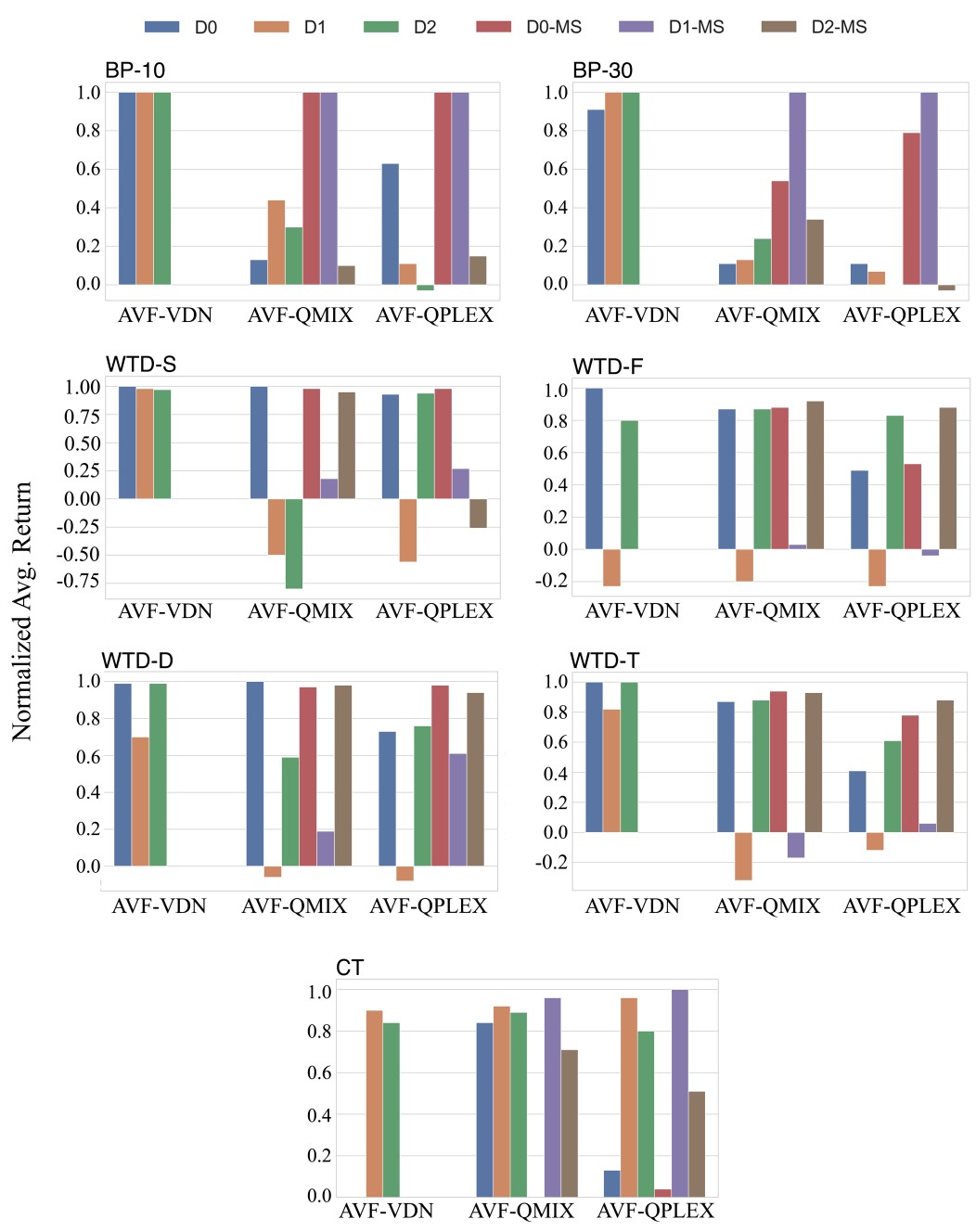

Figure 10: Normalized average return for all our algorithm variations. Tasks have different characteristics affecting the performance of the different update schemes.

In the following, we report all the training curves for the proposed algorithms, omitting the "AVF" prefix for simplicity.

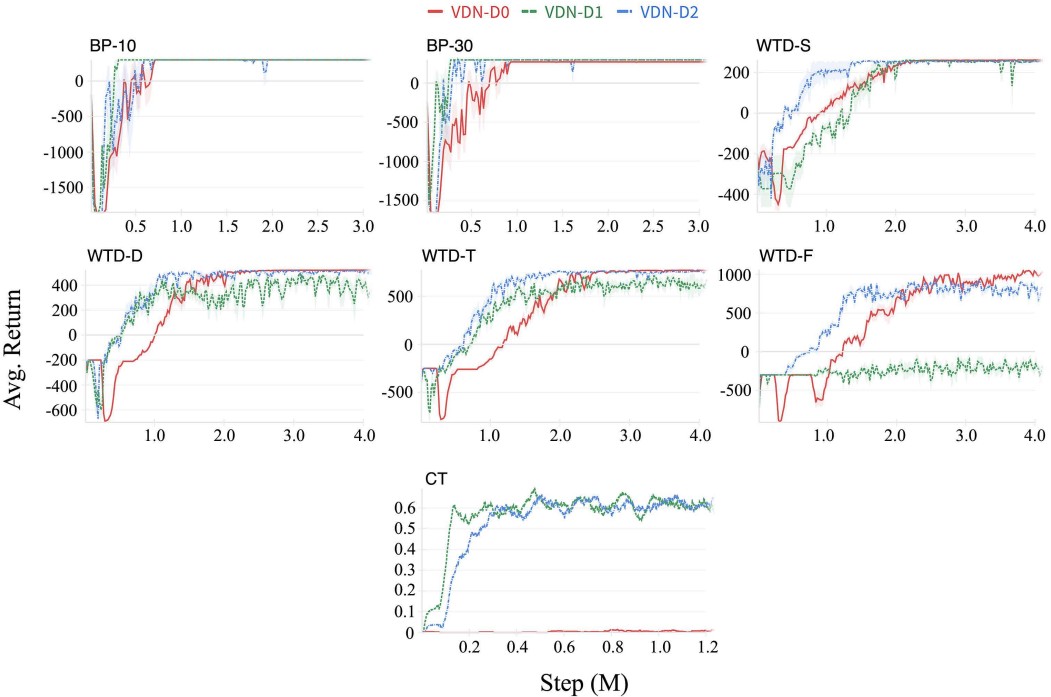

Figure 11: Avg. return over training for AVF-VDN-{D0, D1, D2} using the macro-state.

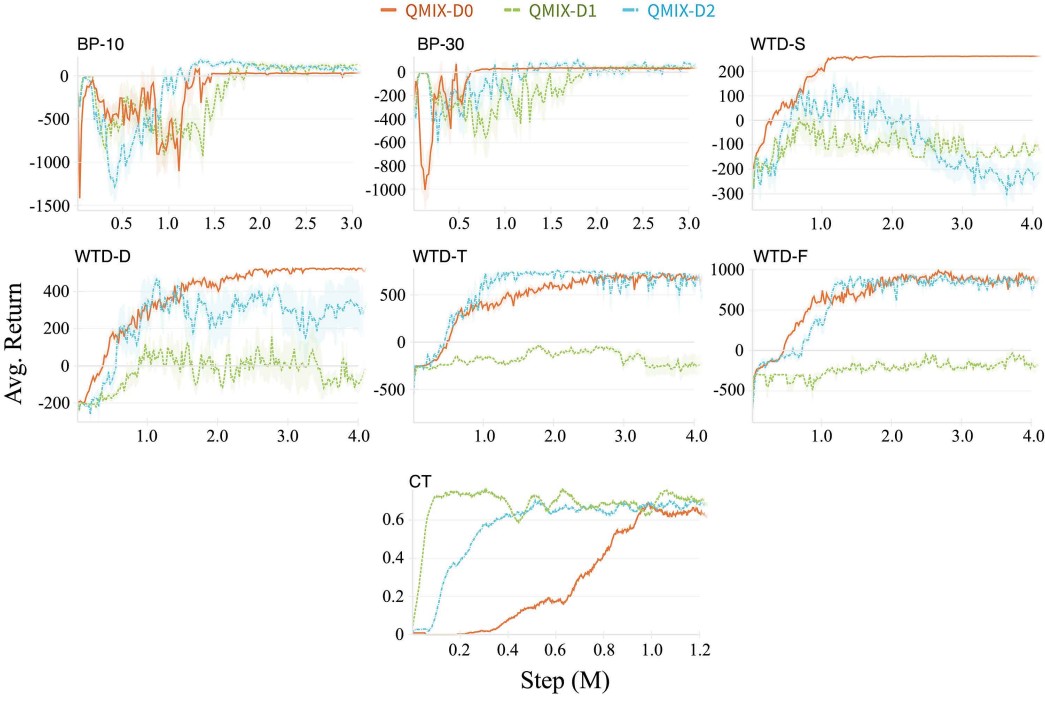

Figure 12: Avg. return over training for AVF-QMIX-{D0, D1, D2} using the macro-state.

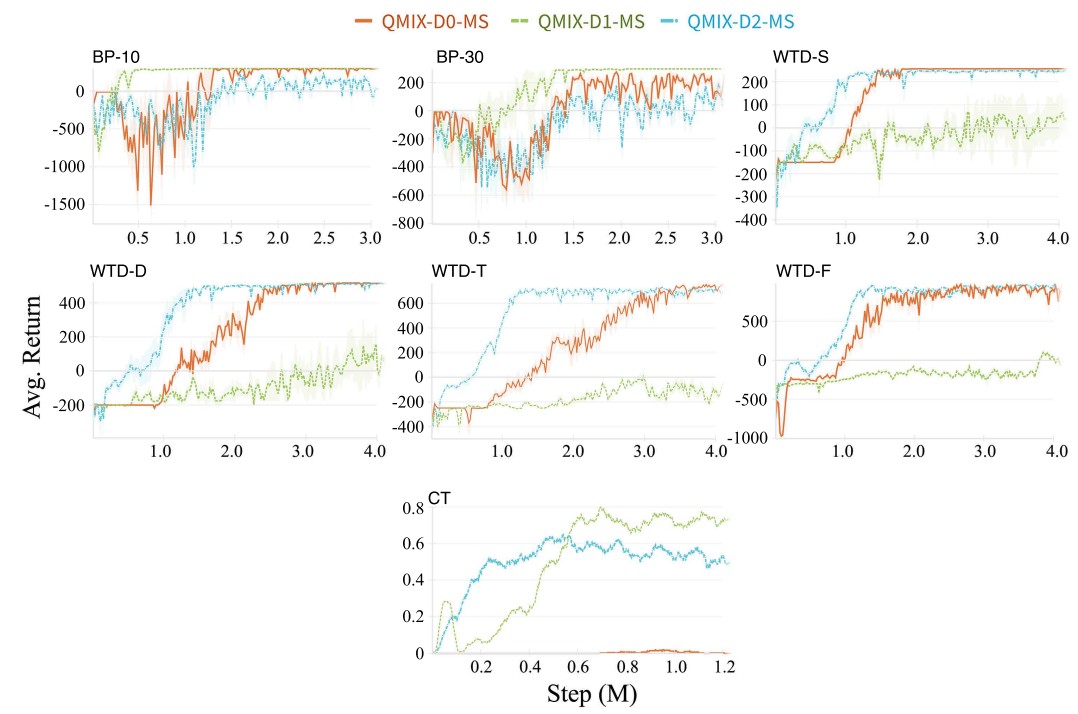

Figure 13: Avg. return over training for AVF-QMIX–{D0, D1, D2}-MS using the joint macro-state.

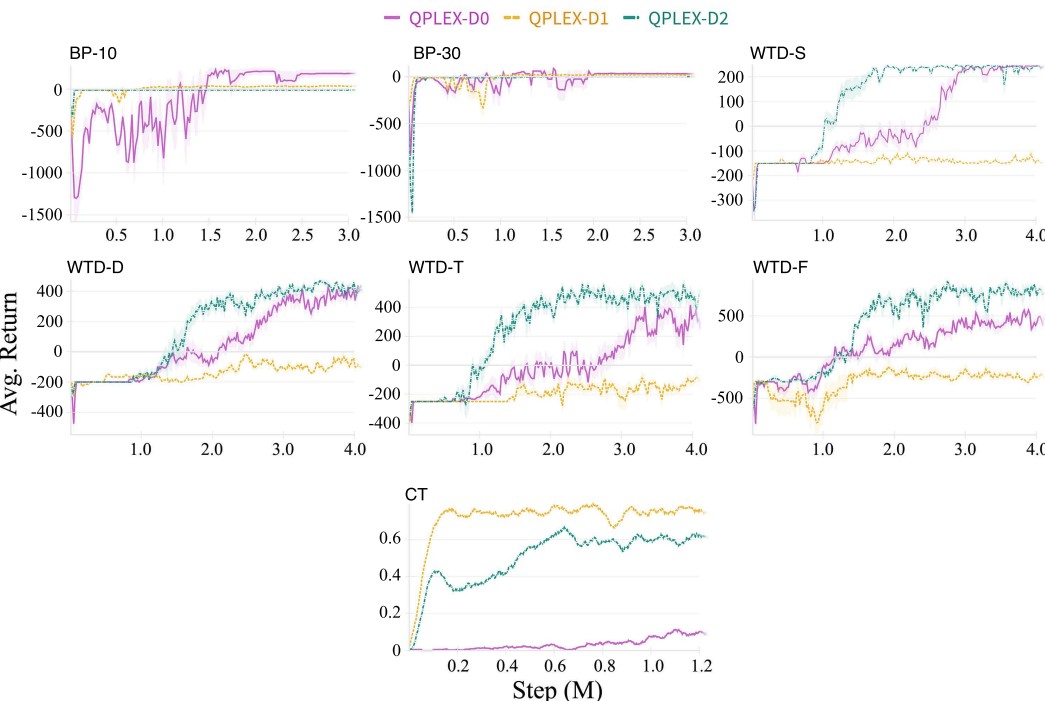

Figure 14: Avg. return over training for AVF-QPLEX-{D0, D1, D2} using the macro-state.

Moreover, Figure 16 shows the training curves for previous macro-action baselines (Xiao et al., 2022; 2020a; Xu et al., 2023).

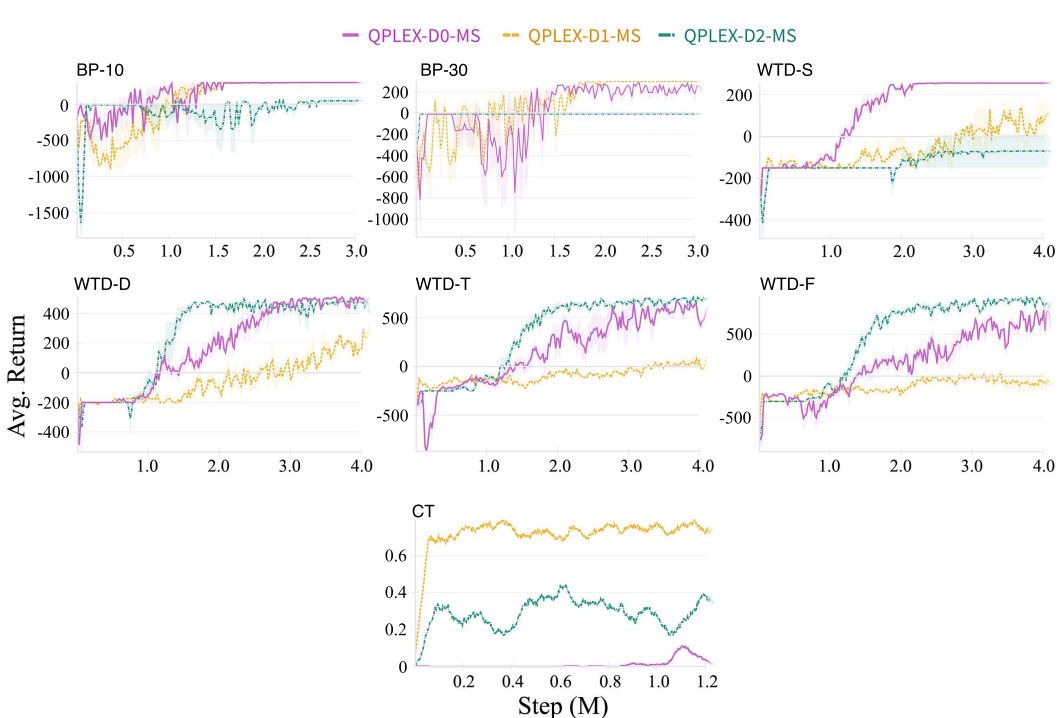

Figure 15: Avg. return over training for AVF-QPLEX-MS-{D0, D1, D2} using the joint macro-state.

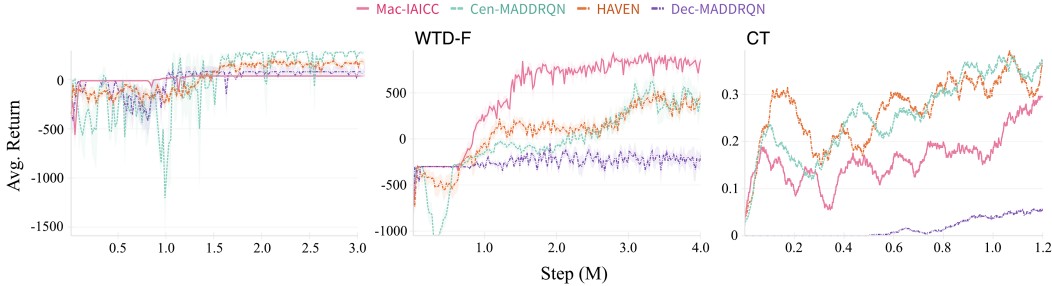

Figure 16: Avg. return over training for Dec-MADDRQN, Cen-MADDRQN, Mac-IAICC, HAVEN.

