# OpenReview forum: "Asynchronous Factorization for Multi-Agent Reinforcement Learning"
_ICLR.cc/2025/Conference — Submitted to ICLR 2025_

### Official Review · Reviewer_iAcy · 2024-10-23

**Soundness:** 3
**Presentation:** 3
**Contribution:** 2
**Rating:** 5
**Confidence:** 4

**Summary:**

This paper investigates an asynchronous multi-agent setting, extending value factorization methods to this context. It introduces Macro-Action-Based IGM and applies relevant value factorization methods within the macro-action framework. The experimental results demonstrate the effectiveness of the proposed methods on macro-action benchmarks.

**Strengths:**

1. The paper addresses an intriguing and underexplored area by applying value factorization methods to agents utilizing macro-actions.

2. The methodology is clearly articulated, supported by figures that enhance understanding.

3. The experimental results show notable improvements on benchmarks.

**Weaknesses:**

- The current method appears straightforward; a deeper exploration is warranted. For instance, the proposed MAC-IGM and MacAdv-IGM lack insights into algorithm design. Since MAC-IGM encompasses a broader class of functions over the primitive IGM, it would be beneficial to investigate more effective factorization or simple modifications tailored for asynchronous tasks rather than merely applying previous methods. As it stands, the current variant of IGM serves mainly as a verification of existing value factorization methods, diminishing its contribution.

- Asynchronous updates require further examination:
  - Why must agents with ongoing macro-actions be masked?
  - Using 0 as the masked value in D2 seems problematic; if all rewards and Q-functions are negative, more agents with ongoing macro-actions would lead to lower rewards assigned for others.
  - For QPLEX, since its mixing network also requires actions as input, will this be masked in D1 and D2?

  A more principled approach to determining asynchronous updates, potentially aligned with the MAC-IGM definition, should be considered.

- The environmental results are unclear:
  - The comparisons in Section 5.1 between D0, D1, D2, w/ and w/o MS are hard to discern from Figure 6.
  - The comparison of AVF to prior methods is vague; using the best performance among VDN, QMIX, QPLEX, and all update and macro-state variants seems unfair.
  - Since the proposed method is value-based, incorporating baselines like IQL with macro-actions would be beneficial.
  - The figures are unclear, and error bars are missing. Utilizing vector graphics instead of raster graphics is recommended, as the latter become blurry when enlarged.
  - How many seeds were used for each experiment?

**Questions:**

Refer to the weaknesses mentioned above.

---

> ### Author Response · Authors · 2024-11-20
> **Response to Reviewer iAcy**
>
> Thank you for your constructive feedback, which has greatly helped us improve the clarity and presentation of our paper. Below, we address the key concerns you raised, with references to the revised manuscript for further details.
>
> > the proposed MAC-IGM and MacAdv-IGM lack insights into algorithm design…
>
> We acknowledge that the connection between Mac-IGM, MacAdv-IGM, and their implementation in asynchronous value factorization (AVF) algorithms could have been better articulated in our original submission.
> The critical link lies in the conditional prediction of macro-action value functions, which enables accurate joint Q-value estimation even when agents asynchronously terminate their macro-actions. Without the conditional operator, Q-value predictions would incorrectly assume that all agents simultaneously initiate new macro-actions, leading to suboptimal performance. This mechanism ensures agents complete their current macro-actions before sampling new behaviors, preventing premature decisions. Thus, conditional Q-value predictions are pivotal for formalizing Mac-IGM and ensuring reliable asynchronous performance.
> All AVF algorithms integrate conditional value function predictions into their architectures and update rules, ensuring compliance with Mac-IGM and MacAdv-IGM principles. To further illustrate these connections, we have added a representative example in Appendix B, demonstrating the full expressiveness of AVF-QPLEX for MacAdv-IGM and bridging the gap between theory and practice.
> Additionally, our work introduces more than just conditional value prediction. We also propose novel detaching schemes for handling macro-actions and identify/address key challenges in utilizing state representations within asynchronous factorization architectures.
>
> > Asynchronous updates require further examination…
>
> We believe the explanation provided above clarifies the importance of masking agents with ongoing macro-actions, as this ensures that updates respect the asynchronous nature of macro-actions. However, we would be happy to provide additional clarification if needed.
> Regarding the use of the 0 mask in D2, we kindly ask the reviewer to elaborate on their statement: “More agents with ongoing macro-actions would lead to lower rewards assigned for others.” The 0 mask ensures that the mixer ignores the value of agents with ongoing macro-actions, preventing these agents from being incorrectly updated during backpropagation.
> For AVF-QPLEX, the mixing network receives information about ongoing macro-actions in both D1 and D2 configurations. We have clarified this detail in the revised manuscript. Overall, D0, D1, and D2 represent three distinct approaches to asynchronous updates, each aligned with the Mac-IGM definition.
>
> > Environmental results are unclear…
>
> We agree that the presentation of results in our original submission could have been improved, and we have made several changes to address this concern:
> - We revised most of Sec. 5 to improve clarity and organization, and we replaced the bar plots in Section 5.1 with clear tabular representations. Full training runs remain available in the appendices.
> - To address fairness concerns, we now present the results of prior methods in a separate table. This allows readers to compare these performances with any AVF method of their choice. This comparison is further discussed in the new Section 5.2.
> - The Dec-MADDRQN method, which we use as a baseline, is an implementation of IQL with macro-actions. This aligns with the reviewer’s suggestion to incorporate baselines like IQL with macro-actions.
> - We have revised all figures in the main paper and supplemental material to enhance their clarity.
> - We also clarified details about our experimental setup. Specifically, we noted that each experiment considers 20 seeds and moved this information to the beginning of Section 5 for greater visibility.
>
> We sincerely thank you once again for your detailed and constructive feedback. We hope our revisions and additional clarifications address your concerns. Please feel free to reach out if you have further questions or suggestions.

---

> > ### Comment · Reviewer_iAcy · 2024-11-22
> >
> > Some follow-up questions:
> >
> > 1. "The critical link lies in the conditional prediction of macro-action value functions." However, conditional value functions should already be known in macro-action settings and do not appear to be a novel contribution of this work. From my understanding, with the awareness of conditional value functions, existing value factorization methods could be straightforwardly adapted into a macro-action version that satisfy Mac-IGM. This makes it unclear how useful Mac-IGM itself truly is.
> >
> > 2. "The 0 mask ensures that the mixer ignores the value of agents with ongoing macro-actions, preventing these agents from being incorrectly updated during backpropagation." However, my concern is that when agents with ongoing macro-actions are masked to 0, the value assignment for other agents without ongoing macro-actions will become incorrect.
> >
> > 3. It is still difficult for me to clearly identify which method generally performs best and whether any method consistently outperforms all baselines. Additionally, I could not find the performance curves of the baseline methods.

---

> > > ### Author Response · Authors · 2024-11-24
> > > **Response to Reviewer iAcy's Comment**
> > >
> > > Thank you for your thoughtful follow-up questions and clarifications. We appreciate your engagement in this discussion and are happy to address the additional concerns you raised:
> > >
> > > >  However, conditional value functions should already be known in macro-action settings and do not appear to be a novel contribution…
> > >
> > > We acknowledge that conditional value functions have been explored in prior work on fully centralized asynchronous macro-action methods, as detailed in the preliminaries (Sec 2.2.1). Our paper does not claim novelty regarding their introduction. Instead, we focus on a crucial aspect. The primitive IGM has been essential for proving the theoretical soundness of primitive factorization algorithms, ensuring consistency between local and joint action selection. Similarly, we need the Mac-IGM (and MacAdv-IGM) to evaluate whether asynchronous macro-action factorization algorithms—including future methods—are principled. As also recognized by Reviewer Qp2t, employing factorization and asynchronous macro-action learning has not been explored before, and we believe our additional insights on the theoretical framework have further improved the clarity of our work. Additionally, our paper bridges a gap in the literature by demonstrating the relationship between these asynchronous and primitive versions.
> > >
> > > > existing value factorization methods could be straightforwardly adapted into a macro-action version that satisfies Mac-IGM.
> > >
> > > While straightforward adaptations of existing methods may satisfy Mac-IGM, we believe this perspective overlooks the other technical contributions of our work. Our proposed macro-state representation and update schemes (D1/D2) focus on the unique challenges of asynchronous factorization. These contributions leverage the asynchronous nature of the problem, resulting in higher performance than unconditioned asynchronous factorization methods and straightforward extensions of existing value factorization approaches. Our evaluations show that simple adaptations rarely outperform AVF algorithms that incorporate these advancements.
> > >
> > > > The 0 mask ensures that the mixer ignores the value of agents with ongoing macro-actions
> > >
> > > We appreciate this clarification and partially agree. To address this, we have updated our manuscript (see footnotes 3 and 4) to highlight scenarios where this masking strategy may encounter practical challenges. Specifically, incorrect estimations could arise if the mixing architecture does not consider the joint macro-history as input, which provides sufficient context to address such an issue. Nevertheless, we note that value masking has demonstrated higher performance than other update schemes in some of the asynchronous macro-action benchmark domains.
> > >
> > > > clearly identify which method generally performs best and whether any method consistently outperforms all baselines.
> > >
> > > Across tasks, methods utilizing the joint macro-state (MS) generally achieve the highest performance, regardless of the update scheme (except AVF-VDN, which does not incorporate this additional information). However, the effectiveness of specific update schemes varies by task, and no single approach consistently outperforms the rest across all settings. Interestingly, AVF-QMIX often outperforms the more complex AVF-QPLEX, likely because its simpler architecture is better suited to the shorter horizons typical of macro-action domains.
> > > The lack of a uniquely better performing algorithm highlights an exciting opportunity for future research as developing novel asynchronous macro-action methods that leverage factorization could further advance the field. Our work provides a foundational framework for this exploration, addressing key challenges unique to asynchronous macro-action-based agents and paving the way for new principled methods.
> > >
> > > > I could not find the performance curves of the baseline methods.
> > >
> > > Thank you for bringing this to our attention! We have added the training curves for macro-action-based baselines to the revised manuscript.
> > >
> > >
> > > We hope our new revisions clarify your concerns. Please feel free to reach out if you have further questions.

---

### Official Review · Reviewer_3SjF · 2024-10-27

**Soundness:** 2
**Presentation:** 2
**Contribution:** 3
**Rating:** 3
**Confidence:** 4

**Summary:**

The paper studies temporal abstraction in MARL, focusing on Asynchronous Value Factorization (AVF). It adjusts the IGM consistency and common value factorization methods to the AVF setting, where agent utilities with ongoing macro-actions are excluded from the gradient calculation. The approaches are evaluated on some small benchmark domains.

**Strengths:**

- Studies a well-motivated but somewhat neglected problem
- Well-written and easy to understand
- The limitations and broader impact are clearly stated

**Weaknesses:**

**Limitation**

- The paper assumes macro-actions to be pre-defined but realistically this does not seem feasible to me: Since macro-actions consist of primitive actions, the macro-action space scales exponentially w.r.t. time, which makes a manual definition of adequate (or even optimal) macro-actions prohibitive.

**Significance**

- I am sceptical about the optimality claim regarding Macro-Dec-POMDPs, which depends on the actual macro-action definition. E.g., considering Dec-Tiger, one could define a macro-action that only consists of Listening actions (without opening a door), which would never yield the optimal reward - regardless of what the other agents does.
- As far as I understood, the most significant technical contribution is only to exclude the agent utilities of ongoing macro-actions from the gradient calculation, e.g., using masking
- The experimental is mainly a self-comparison, i.e., ablations, without including any baseline known from hierarchical MARL [1] or temporal abstraction in MARL [2], which also consider macro-action-based value factorization

**Presentation**

- In Section 4, the first and second paragraph overlap visually which looks weird.
- The labels of Figures 3, 4, 5, 6, and 8 are too small (they are unreadable when printed). Even worse, the plots Figures 5, 6, 7, and 8 pixelate when zooming in, which does not help readability either. Thus, I am uncertain about the significance and quality of the results.

**Literature**

[1] Xu et al, "HAVEN: Hierarchical Cooperative Multi-Agent Reinforcement Learning with Dual Coordination Mechanism", AAAI 2023

[2] Tang et al., "Hierarchical Deep Multiagent Reinforcement Learning with Temporal Abstraction", 2018

**Questions:**

None

---

> ### Author Response · Authors · 2024-11-20
> **Response to Reviewer 3SjF**
>
> We appreciate your review and the opportunity to improve our work. We have carefully revised our paper to address your concerns and provide further clarity on the motivation and contributions of the work. Below, we respond to each of your points, highlighting the revisions and providing references to the updated manuscript.
>
> *Presentation*
>
> We acknowledge that the clarity of the figures in our original submission could have been improved. To address this, we have revised all figures in both the main paper and the supplemental material. These updates include increased label sizes, improved plot quality, enhanced readability and structure. Additionally, we replaced the original bar plots, which may have caused your uncertainty about result quality, with a tabular format for clearer comparison. Full training runs remain available in the appendices.
>
> *Significance*
>
> > I am sceptical about the optimality claim regarding Macro-Dec-POMDPs
>
> Thank you for pointing this out. To clarify, we claim optimality with respect to the given macro-actions. This is similar to the primitive action case where algorithms can be optimal with respect to the given (primitive) actions. As discussed in most of the works we refer to in Section 2.2 and 2.2.1, the trade-off between optimality and empirical performance in Macro-Dec-POMDP-based algorithms is well established in the literature.
>
> > the most significant technical contribution is only to exclude the agent utilities of ongoing macro-actions
>
> We respectfully disagree with this interpretation of our contributions. While it is true that our work introduces novel detaching schemes for handling macro-actions, it also:
> - Identifies and addresses challenges in using the state for asynchronous factorization architectures.
> - Formalizes the underlying theory for Macro-Dec-POMDP factorization methods.
> - Bridges this theory with practical algorithms. We encourage the reviewer to refer to our detailed response to Reviewer Qp2t for a comprehensive discussion of this point.
>
> > The experimental is mainly a self-comparison…
>
> We would like to clarify that our experimental evaluation goes beyond self-comparison. In addition to extensive ablation studies that validate the design and technical contributions of our asynchronous value factorization (AVF) algorithms, we compare against existing asynchronous MARL baselines, including both value-based and policy-gradient methods (e.g., Mac-IAICC, Dec-MADDRQN, and Cen-MADDRQN).
> Regarding your suggestion to compare with [1, 2], while we appreciate the recommendation, we note these works are not closely related to our framework. As also discussed in [1] and Xiao et al. (2022), these studies do not address asynchronicity, as they assume all agents execute macro-actions of the same duration. Thus, they are more aligned with synchronous MARL approaches (Section 2.2.1).
> Nonetheless, to address your request for broader comparisons, we have included results for HAVEN [2] in the updated manuscript (Table 2). Using the same hyperparameters as AVF-QMIX-D0 and the best macro-action duration identified in a preliminary sweep (5 steps), HAVEN demonstrates lower performance compared to all asynchronous MARL baselines and AVF algorithms. Please refer to Section 5.2 for further details.
>
> *Limitation*
>
> > The paper assumes macro-actions to be pre-defined but realistically this does not seem feasible to me
>
> We agree that the assumption of predefined macro-actions represents a limitation of our work, as we also discuss in the paper. However, we note that in RL, people assume they are given an action space also in the primitive case. Macro-actions are more general, allowing us to consider actions with different durations—a standard assumption in all previously published asynchronous MARL studies (in fact, we adopt existing benchmark environments that provide these predefined macro-actions). Asynchronous settings are common in the real world but have been rarely studied in the literature. For this reason, principled methods are needed for the MacDec-POMDP case before extending them to learn macro-actions. Hence, as highlighted in our paper, learning asynchronous macro-actions is indeed an exciting direction for future work. However, our primary focus is to establish the theoretical framework and scalable asynchronous MARL algorithms, laying the foundation for such advancements.
>
> We are grateful for your constructive feedback, which has helped us improve the clarity and rigor of our work. We hope these clarifications address your concerns and demonstrate the significance of our contributions. Please do not hesitate to reach out with further questions or suggestions.

---

### Official Review · Reviewer_Qp2t · 2024-11-03

**Soundness:** 3
**Presentation:** 2
**Contribution:** 2
**Rating:** 6
**Confidence:** 3

**Summary:**

This paper proposes a method to effectively handle macro-actions in multi-agent reinforcement learning (MARL) through an asynchronous learning structure. To address the inefficiencies and temporal inconsistencies that arise when agents have different action durations in synchronous learning frameworks, the paper introduces the Asynchronous Value Factorization (AVF) method. This approach allows agents to update independently in environments involving macro-actions, improving learning efficiency and scalability. The paper provides theoretical foundations through the concepts of Macro-IGM and MacAdv-IGM, demonstrating that macro-action-based learning has a broader expressiveness compared to traditional single-action methods. Experimental validation shows that AVF outperforms synchronous methods in various macro-action scenarios, achieving superior performance in asynchronous policy learning.

**Strengths:**

- Solving practical issues. The paper introduces an asynchronous learning structure that addresses the temporal inconsistencies and update delays inherent in synchronous approaches. This enables agents to update independently, significantly improving learning efficiency and environmental adaptability. This design effectively reflects the real-world asynchronous nature of multi-agent systems, solving practical issues.
- Theoretical Extensibility: The concept of Macro-IGM provides greater representational power for handling macro-actions, ensuring that local agent actions remain consistent with the global policy. This allows for stable policy optimization even in asynchronous settings, demonstrating an expanded expressiveness and applicability beyond traditional methods.
- Compatible with Existing Algorithms: The proposed asynchronous update mechanism can be integrated with existing algorithms like QMIX and QPLEX, enhancing their performance by addressing temporal inconsistencies and improving efficiency. This compatibility allows the AVF method to be applied broadly across various MARL frameworks, increasing its practical utility.

**Weaknesses:**

- Theoretical Analysis Limitations: Although the paper presents theoretical propositions such as Macro-IGM and MacAdv-IGM, their practical relevance to the proposed AVF algorithm is limited. The theoretical framework is broad and lacks direct applicability, as it does not clearly define or demonstrate how the function class  F  satisfying these conditions is implemented within the algorithm. Additionally, the propositions do not significantly bridge the gap between theoretical findings and practical performance, raising questions about the real impact of these proofs on the algorithm’s effectiveness.
- The paper’s asynchronous learning structure and handling of macro-actions, while effective, are relatively straightforward and have been explored in MARL research before. This makes the contribution seem incremental rather than a significant innovation, potentially limiting its perceived impact in advancing the field.

**Questions:**

Could you explain how the theoretical propositions, specifically Macro-IGM and MacAdv-IGM, concretely impact the learning performance and stability of the AVF algorithm? I would like clarification on how these theoretical proofs contribute directly to the design and performance improvement of the algorithm.

---

> ### Author Response · Authors · 2024-11-20
> **Response to Reviewer Qp2t**
>
> We appreciate your thoughtful review and valuable suggestions. Your comments have significantly helped us improve the clarity and quality of our paper. Below, we address the concerns you raised, with references to the revised paper for further details.
>
> > The theoretical framework … does not clearly define or demonstrate how the function class F satisfying these conditions is implemented within the algorithm. … the propositions do not significantly bridge the gap between theoretical findings and practical performance ….
>
> We acknowledge that the original manuscript could have better explained the connection between macro-action-based IGM principles and their implementation in asynchronous value factorization (AVF) algorithms. Our efforts to keep the paper concise may have inadvertently hindered clarity in this regard.
> The critical link between the theoretical framework and the algorithms lies in the conditional prediction of macro-action value functions. These predictions enable accurate estimation of the joint Q-value, even when agents asynchronously terminate their macro-actions at different steps. Without the conditional operator, the Q-value estimation would inaccurately assume that all agents simultaneously initiate new macro-actions, leading to suboptimal performance. This mechanism ensures agents do not prematurely sample new high-level behaviors before completing their current macro-actions. Overall, in asynchronous settings, where macro-action durations are typically unknown and heterogeneous, maintaining consistency between joint and local macro-action value functions is essential for principled factorization. Conditional Q-value predictions are thus pivotal for correctly formalizing Mac-IGM and achieving reliable asynchronous performance. All AVF algorithms incorporate conditional value function predictions into their architectures and update rules, ensuring adherence to Mac-IGM and MacAdv-IGM principles. These clarifications are detailed in the revised manuscript, specifically from line 167, 179, and 305.
> Additionally, we have included a representative example in Appendix B, which demonstrates the full expressiveness of AVF-QPLEX for MacAdv-IGM, further bridging the gap between theory and practice.
>
> > The paper’s asynchronous learning structure and handling of macro-actions, while effective, are relatively straightforward and have been explored in MARL research before.
>
> We kindly ask the reviewer to clarify this claim. While it is true that macro-actions and asynchronous learning have been studied in multi-agent reinforcement learning (MARL), our work addresses a critical gap by presenting principled centralized training with decentralized execution (CTDE) algorithms specifically tailored for asynchronous MARL. To our knowledge, this is the first work to do so. Our extensive set of experiments empirically demonstrates the practical advantages of the CTDE framework, highlighting the value of principled algorithms in asynchronous MARL scenarios. We believe our results will inspire further research in this area.
>
> Thank you once again for your constructive feedback. We hope these clarifications address your concerns, and we are happy to discuss any additional questions that may arise during this review process.

---

> > ### Comment · Reviewer_Qp2t · 2024-11-24
> >
> > Thank you for your detailed and thoughtful response.
> >
> > I meant "asynchronous learning structure" and "handling of macro-actions" have been individually studied. I agree that it has not been explored in this way before.
> >
> > Your clarification and modification regarding the theoretical framework have improved the clarity of the paper. Taking this into consideration, I will adjust my score accordingly.

---

### Meta-Review · Area_Chair_7jPi · 2024-12-20

**Metareview:**

This paper proposes Asynchronous Value Factorization (AVF), a new method for handling macro-actions (temporally extended actions) in multi-agent reinforcement learning (MARL). AVF addresses the inefficiencies and inconsistencies of synchronous learning when agents have different action durations by allowing them to update independently.

Strengths
-----------
- **Practical relevance:** AVF tackles a real-world issue in MARL where agents often operate asynchronously with varying action durations.

- **Improved efficiency and scalability:**  By enabling independent updates, AVF enhances learning efficiency and scalability in macro-action scenarios.

- **Theoretical foundation:** The paper introduces Macro-IGM and MacAdv-IGM, theoretical concepts that extend the expressiveness of MARL with macro-actions.

- **Compatibility:** AVF can be integrated with existing MARL algorithms like QMIX and QPLEX, enhancing their performance in asynchronous settings.

Weaknesses
--------------
- **Limited theoretical significance:** While the paper presents theoretical propositions, their practical connection to the AVF algorithm remains unclear. The theory lacks direct applicability and doesn't clearly demonstrate how it contributes to the algorithm's effectiveness.

- **Incremental contribution:** The asynchronous learning structure and handling of macro-actions, while effective, are considered relatively straightforward and lack significant novelty.

- **Concerns on experiments:** Reviewers raised questions about the clarity of the experimental results, the choice of baselines, and the need for more detailed comparisons with existing methods in hierarchical MARL.

AVF offers a practical solution for dealing with macro-actions in MARL, demonstrating improved efficiency and scalability in asynchronous settings. However, the paper's theoretical contribution needs further clarification and its novelty is considered somewhat limited.

**Additional Comments On Reviewer Discussion:**

Concerns remained about the contribution of the paper and the relevance of presented results, in particular about the theoretical claims about Macro-IGM and MacAdv-IGM.

---

### Decision · Program_Chairs · 2025-01-22

Reject